

# Validation of Aeolus wind profiles using ground-based lidar and radiosonde observations at La Réunion Island and the Observatoire de Haute Provence

Mathieu Ratynski[1], Sergey Khaykin[1], Alain Hauchecorne[1], Robin Wing[1,a], Jean-Pierre Cammas[2], Yann Hello[2], Philippe Keckhut[1]

[1]Laboratoire Atmosphères, Observations Spatiales (LATMOS), UVSQ, Sorbonne Université, CNRS, IPSL, Guyancourt, France
[2]Observatoire des Sciences de l'Univers à La Réunion (OSU-R, UAR3365), Saint-Denis, La Réunion
[a] now at: Leibniz-Institute for Atmospheric Physics, Kühlungsborn, Germany

*Correspondence to:* Mathieu Ratynski (mathieu.ratynski@latmos.ipsl.com)

**Abstract.** European Space Agency's (ESA) Aeolus satellite mission is the first Doppler wind lidar in space, operating in orbit for more than three years since August 2018 and providing global wind profiling throughout the entire troposphere and the lower stratosphere. The Observatoire de Haute Provence (OHP) in southern France and the Observatoire de Physique de l'Atmosphère à La Réunion (OPAR) are equipped with ground-based Doppler Rayleigh-Mie lidars, which operate on similar principles to the Aeolus lidar, and are among essential instruments within ESA Aeolus Cal/Val program. This study presents the validation results of the L2B Rayleigh-clear HLOS winds from September 2018 to January 2022. The point-by-point validation exercise relies on a series of validation campaigns at both observatories: AboVE (Aeolus Validation Experiment) that were held in September 2019 and June 2021 at OPAR, and in January 2019 and December 2021 at OHP. The campaigns involved time-coordinated lidar acquisitions and radiosonde ascents collocated with the nearest Aeolus overpasses. During AboVE-2, Aeolus was operated in a campaign mode with an extended range bin setting allowing inter-comparisons up to 28.7 km. We show that this setting suffers from larger random error in the uppermost bins, exceeding the estimated error, due to lack of backscatter at high altitudes. To evaluate the long-term evolution in Aeolus wind product quality, twice-daily routine Météo-France radiosondes and regular lidar observations were used at both sites. This study evaluates the long-term evolution of the satellite performance along with punctual collocation analyses. On average, we find a systematic error (bias) of -0.92 ms$^{-1}$ and -0.79 ms$^{-1}$ and a random error (scaled MAD) of 6.49 ms$^{-1}$ and 5.37 ms$^{-1}$ for lidar and radiosondes, respectively.

## 1 Introduction

Wind velocity is one of the fundamental meteorological variables describing the atmospheric state. Assimilating atmospheric wind observations into numerical weather prediction (NWP) models is crucial to understand the evolution and structure of weather dynamics, air quality monitoring, forecasting, and climate and meteorological studies. Accurate NWPs are essential





for commercial activities such as agriculture, fisheries, construction, transportation, energy development, and daily life. Therefore, continuous global wind profiling crucial to our knowledge of atmospheric dynamics and improved accuracy in numerical weather predictions (Houchi et al., 2010; Albertema et al., 2019; Stoffelen et al., 2005; 2020).

The wind measurements are conducted with a large variety of techniques : Radiosondes (Houchi et al., 2010), Wind Profiler Radars (WPRs) (Rogers et al., 1993), Ground-based Doppler Wind Lidars (DWLs) (Chanin et al., 1989; Baumgarten et al., 2010; Xia et al., 2012), Sodars (Anderson et al., 2005), Metal Resonance Lidars (She et al., 2004), Microwave Radiometers (Rüfenacht et al., 2012), Infrasound (Le Pichon et al., 2005), Aircrafts and Airborne Lidars (Prudden et al., 2018, Yan et al., 2015, Lux et al., 2020a), Satellites (Eyre., 2019) and Atmospheric Motion Vectors (Forsythe., 2007). The DWL technique offers a broad altitude range from the lower troposphere up to the lower mesosphere and a vertical resolution of 40 around 100 meters. In that sense, they overcome most of the other instruments in terms of both vertical resolution and altitude range, making them a perfect choice as reference instruments for satellite validation. Unfortunately, there are only a handful of them being operated today.

On August 22, 2018, The European Space Agency (ESA) launched the Aeolus satellite as part of the Living Planet Program. With an initial estimated lifetime of three years, this mission is expected to pave the way for future operational meteorological 45 satellites dedicated to observing the atmospheric wind field in order to advance the understanding of climate processes and atmosphere dynamics (ESA, 2019; Straume et al., 2020). Aeolus is a polar-orbiting satellite in a sun-synchronous dawn-to-dusk orbit at about 320 km altitude. Within seven days, the satellite covers nearly the whole globe. The satellite's payload consists of only one large instrument, a Doppler wind lidar called ALADIN (Atmospheric LAser Doppler INstrument), which is the first-ever Doppler-Rayleigh Wind Lidar (DWL) in space (Stoffelen et al., 2005; Reitebuch, 2012; Kanitz et al., 2019a). 50 ALADIN is a direct-detection high spectral resolution wind lidar providing vertical profiles of the Line-of-Sight (LOS) wind velocity profiles at an angle of 35∘ off-nadir from the ground up to 30 km. While 30 km is the initially stated nominal range, it is very rarely used, favoring smaller ranges of observation. However, as with all satellite sensors, the remote sensing technology and retrieval algorithm require a careful assessment of the quality and validity of the generated data products. Although corrections to several substantial bias sources in the Aeolus L2B winds have been implemented in the data processing 55 (Rennie et al., 2021 Reitebuch et al., 2020; Weiler et al., 2021b), a direct validation against high resolution measurements is required to identify any residual biases in Aeolus's L2B winds.

Aeolus Validation Experiment (AboVE) is a part of the French contribution to the Aeolus cal/val programme, which included a series of intensive measurement campaigns at mid-latitude and tropical stations: Observatoire de Haute Provence (OHP) and at high-altitude Maido observatory at La Reunion Island, OPAR (Observatoire de Physique de l'Atmosphère à La 60 Réunion). The OHP has been an historical place for lidar observations of the atmosphere for several decades (Hauchecorne et al., 1980; Hauchecorne et al., 1991), while the Maido observatory saw its first observation campaigns in 2013 (Keckhut 2015). Both stations are equipped with an extensive suite of lidar instruments, including Rayleigh-Mie Doppler lidars for wind profiling up to 70 km altitude using a double-edge Fabry-Perot interferometer (direct detection method). This technique, pioneered by French Service d'Aeronomie in 1989 (Chanin et al., 1989), is implemented in the Rayleigh channel of the Aeolus



ALADIN instrument (Schillinger et al., 2003), which is why the OHP and OPAR sites are considered to be important contributors to the Aeolus cal/val programme. Both AboVE-OHP and AboVE-Maido cal/val campaigns involved time-coordinated radiosounding measurements during the satellite overpass.

The paper is organized as follows: In Sect. 2, the radiosondes, the ground-based lidars and the designs and operation principles of ALADIN are briefly described with a focus on the commonalities and differences between the two wind lidar
instruments and M10 radiosondes. In Sect. 3, an overview of the two validation campaigns is given, and the procedure of matching the different resolutions of the used data sets is explained. An extensive statistical study is then proposed in Sect. 4 through a statistical comparison of entire datasets over the different regions and time periods, and a long-term validation over both observatories. Additionally, single case studies are presented to give further insight into the error sources. Finally, the results are discussed in Sect. 5, followed by the conclusions in Sect. 6

## 2 Instruments and methods

### 2.1 Experimental Setup

#### 2.1.1 Ground-based lidars

The Observatoire de Haute-Provence (OHP), located in southern France (43.9°N, 5.7° E; 683 m above sea level), is one of the Alpine stations of the Network for Detection of Atmospheric Composition Change (NDACC). The LIOvent Doppler lidar
was deployed at OHP in 1993 and provided the first lidar-based wind climatology in the middle atmosphere (Souprayen et al., 1999). The lidar senses the horizontal wind components by measuring the Doppler shift between emitted and back scattered laser light using a double-edge Fabry-Perot interferometer (Chanin et al., 1989), which is the direct-detection technique implemented in ALADIN's Rayleigh channel. The Doppler shift corresponds to the projection of the horizontal wind components onto the line-of-sight of the laser, inclined off-zenith.

The lidar makes use of a Quanta-Ray Pro290 Q-switched, injection-seeded Nd:YAG laser emitting at 532 nm with a repetition rate of 30 Hz, 800 mJ per pulse energy. The laser beam is cycled successively between three lines of sight with a cadence of 1-2-2 minutes for measuring the zonal and meridional wind components, whereas the vertical pointing is used for calibration. The OHP system includes three fixed telescope subassemblies, each comprising a mosaic of four 50 cm mirrors: one is pointed toward the zenith, while the others are tilted at 40° off the zenith to the North and East directions. The laser
beam is steered by a galvanometric scanner mirror with the three predefined positions. Measurements are limited to nighttime or twilight conditions and the absence of optically thick clouds. After a series of technical upgrades that started in 2013, the LIOvent lidar has become an operational instrument with a capacity of wind profiling up to 75 km (Khaykin et al., 2020) and was approved as a climate monitoring instrument by NDACC in 2021.

In 2012, another Doppler lidar (LiWind) was deployed at the new tropical high-altitude Maido observatory of OPAR (21°S,
55°E, 2200 m a.s.l.) on the island of La Réunion. The LiWind system at OPAR uses the same laser, detection and acquisition





systems as the OHP wind lidar but features a more compact design for the receiver assembly. The telescope is made up of a single rotating 60 cm mirror, which serves for both the emission and reception (Khaykin et al., 2018). Compared to the OHP's lidar, the smaller collective area of LiWind instrument is compensated by the station's higher elevation and the cleaner atmosphere above the Indian Ocean. Both OHP and OPAR lidars provide wind measurements with an accuracy of better than 100 1 ms$^{-1}$ within the entire range of Aeolus altitude coverage. Both instruments will be referred to as ground-based lidar hereinafter.

### 2.1.2 Radiosondes

The radiosonde (RS) wind measurements, based on a simple GPS tracking of the balloon position offer high accuracy and vertical resolution and their inherent errors (e.g., instrument errors) are minor compared to satellite instrument errors. They 105 are well suited to serve as a baseline dataset for the actual atmospheric state to validate the Aeolus HLOS winds. Radiosondes measurements are known to provide a solid reference against which other measurements can be validated (B. Sun et al., 2010; Krisch et al., 2017). Furthermore, radiosondes also provide guidance for observational strategies and requirements when collecting feedback from past collocations campaigns with similar instrumentation (Iwai et al., 2021; Baars et al., 2020; Martin et al., 2021). For each launch, it can be assumed that the observation errors are not correlated 110 between the different radiosondes. It should be noted that RS have the problem of time and space drift in measuring the vertical wind profile (Baars et al., 2020; Martin et al., 2021). The speed and direction of the horizontal wind are calculated using GPS position changes based on the Global Climate Observing System Reference. According to the GCOS Reference Upper-Air Network (GRUAN), the horizontal measurement of wind speed and direction uncertainties are assumed to be between 0.4 and 1 m s$^{-1}$ for the wind velocity and 1∘ for the wind direction (Dirksen et al., 2014). While lidars sample air 115 advected into the fixed lines of sight, radiosonde measurements are made within the local flow. Although these instruments might not sample the exact same volume of air, their measurements are proven to be highly correlated above 500 m (Kumer et al., 2014; Khaykin et al., 2020), which make the radiosondes fully suitable for validation of ground- and space-based lidars.

The RS at OHP and Maido sites were flown under Totex 1200 gr balloons, drifting on average 160 km for OHP and 120 120 km for Maido. These values are considered when computing the spatial offset (also referred to as distance to collocation) between the Aeolus and measurements, defining collocation criteria for comparisons of Aeolus and RS measurements. In this study, the RS measurements were performed with no specific spatial offset criterion in order to assess the impact of the distance to collocation on the bias and random error. The closest distance to collocation was 23 km, and the farthest at 241 km.

### 2.2 Aeolus ALADIN instrument

The payload of Aeolus satellite consists of a single instrument – the Atmospheric LAser Doppler INstrument. The instrument samples the atmosphere with a laser pulse and measures the resulting Doppler shift on the returned signal, resulting from the



different backscatters throughout the different layers of the atmosphere. The frequency shift is caused by the relative motion of the detected elements along the sensor's line of sight. This motion is correlated to the mean wind in the observed volume.

The measurement volume is determined by the vertical resolution, the width of the laser footprint and the ground integration length. The measurements are repeated every 80 kilometers. Each profile comprises several measurements clustered through grouping identifiers (De Kloe et al., 2016). The measurements are approximately 2.85 km (horizontal scale) apart from each other, and each of them is separately analyzed for atmospheric scene classification (Rennie and Isaksen, 2020). The along-orbit interval between individual profiles, obtained by aggregating 30 measurements to improve the signal-to-noise ratio, is

~87 km. The measurements are classified using particle backscatter coefficients or feature lookup algorithm as criteria (Rennie et al., 2020). The Level 2B product consists of four distinct wind observation types, selected using the atmospheric classification performed in the processor chain (Rennie et al., 2020). The method currently applied by ESA is to use the scattering ratio, which is determined as part of Level 1B (L1B) processing (Reitebuch et al., 2014) and used as input for L2B processing (de Kloe et al., 2016; Rennie et al., 2020). For this purpose, a predefined scattering ratio threshold as a function of

height is used. If the scattering ratio is greater than the threshold, the particle scattering is dominant. Under the threshold, only molecular scattering is assumed. Range bins allocated to the same classification type are accumulated in the corresponding observations. The four wind types consist of Rayleigh and Mie-derived winds and can be either categorized clear or cloudy. The Rayleigh and Mie wind retrieval algorithms are applied to their respective two classes of observations.

This paper will only focus on the Rayleigh clear data analysis. Rayleigh clear stands for clear skies. According to the

Rayleigh approach, the winds are measured in regions showing absence of strong Mie backscatter. In the presence of a high backscatter ratio would qualify the data as Rayleigh cloudy, as for cloudy/particle-loaded skies. Rayleigh cloudy products can also provide usable wind measurements. However, contamination of the Mie scattering must first be corrected, which is still in the experimental stage and is not within the scope of this study, therefore limiting our study only to the Rayleigh clear product. The Level 2B product provides an HLOS error estimation for each range bin in the observation profiles. The validity

flag (de Kloe et al., 2016) ensures the validity of the products.

We also apply the quality control guidance L2B threshold from the Aeolus NWP Impact Experiments (Rennie and Isaksen, 2020), except we do not apply any HLOS error threshold. The reason for this choice is to allow for more data to be collocated and also to observe the satellite's behavior in the higher altitudes, where is it shown to have a higher error. The data presented in the following study are from baselines ranging from 2B02 and from 2B11 to 2B13 (September 2018 – January 2022).

During Aeolus commissioning phase, it was noted that ACCDs (accumulation charge-coupled devices) pixels with an increased dark current were present in the memory zone of both ACCDs in the detector unit of ALADIN (Reitebuch et al., 2020; Kanitz et al., 2019b). These pixels are called hot pixels, and their increased dark current can have a time-variable magnitude. The results presented by Weiler et al. (2019a) revealed that by May 2020 6% of ACCD pixels could be classified as hot pixels. Approximately 13% of pixels will be concerned by this issue at the end of the mission's extended life in

November 2022, assuming the hot pixel generation rate does not change (Weiler et al., 2019a). Meanwhile, a hot pixel correction has been in place for Aeolus data since June 14, 2019. Keeping the hot pixel appearance rate under a minimum is



of the utmost importance: it can greatly affect the collocations between Aeolus and the reference instruments, making them harder to estimate the nominal behavior of the satellite (See Sect. 4.1).

The wind is observed orthogonal to the satellite ground track, pointing 35º off-nadir, away from the Sun. (H)LOS means (horizontal) line of sight. A single wind component, called vLOS, is measured along the satellite's line-of-sight (LOS). The latter is then converted into the HLOS wind speed, by assuming that the vertical wind speed $w$ is negligibly small. Equations 1 and 2 allow obtaining the vLOS and HLOS based on the three cartesian wind components $u$ (zonal wind), $v$ (meridional wind), and w (vertical wind). If $w$ is assumed to be minor, the difference between vLOS and HLOS becomes negligible. The angle Ψ represents the elevation of the target-to-satellite pointing vector (55°) and the angle θ is the topocentric azimuth of the

target-to-satellite pointing vector, measured clockwise from north.

$$v_{LOS} = v_{HLOS} \cos(\Psi) + w \sin(\Psi) \quad (1)$$

$$v_{HLOS} = -u \sin(\theta) - v \cos(\theta) \quad (2)$$

**2.3 Comparison setup**

**2.3.1 Adaptation of the measurement grid**

Two significant aspects must be considered for adequate comparison of the radiosonde wind profiles with the Aeolus wind data. First, the two instruments' different horizontal and vertical resolutions necessitate an adaptation of the radiosondes' measurement grid to that of Aeolus. Aeolus' data format consists of 24 vertical range bins that divide the atmosphere, resulting

in wind profiles that can be obtained between 0 and 30 km, displaying a vertical resolution between 250 and 2 km (Reitebuch et al., 2014). The distribution of these 24 range bins is defined through a dedicated Range Bin Settings (RBS). Multiple RBS settings are activated at the same time; The main reason for adding or changing a RBS is to address a specific need, such as better sampling at specific heights. The RBS can therefore vary depending on the latitude and the time, which is all adjusted operationally. In this study, we operate with the bins of varying sizes (from 500 m to 1500 m) and a vertical range from the

ground up to a maximum altitude (varying between 17.8 km to 28.7 km). The radiosondes data have a vertical range up to 35 km, depending on the balloon burst altitude, and the vertical resolution is approximately 5 m at the typical rate of climb of 5 m s–1. The ground-based lidars have a vertical range of up to 75 km and a resolution of 120 m.

Since the instruments have around 5000 measurement points for the radiosondes down to 200 measurements points for the ground-based lidars, compared to the satellite's 24 (at best, if all the measurements have passed quality checks) measurements

for a single profile, a down sampling of these two reference datasets is required. Each Aeolus profile is used as a reference for the collocated profiles interpolations, meaning that the interpolation grid is specific to each satellite observation. The downsampling begins with an averaging of the reference measurements between the middle points of the reference bins, and





then an interpolation of these values to their corresponding reference bin. The result will be a reference profile downsampled to the corresponding numbers of Aeolus bins present in the profile (~21 on average, when considering the validity flags).

### 2.3.2 Consideration of the different viewing geometries

The second significant aspect to consider when comparing different instruments is the different viewing geometries. Because Aeolus only measures in one direction, it is necessary for the other two instruments, the ground-based lidars, and the radiosondes, to project their measurements onto the same line of sight. The HLOS wind component is computed as a linear function of the zonal wind component $u$ and the meridional wind component $v$ using Eq.2. Where $\theta$ (259.9° for OHP and 259.0° for Maido) is the topocentric azimuth angle, which is defined clockwise from north of the horizontal projection of the target to the satellite pointing vector. Therefore, each observation site has its own azimuth angle value.

### 2.3.3 Statistical terms and methods

The offset between Aeolus and the reference data, also referred to as bias, representing the systematic error of the Aeolus wind measurements, is studied alongside the scaled Median Absolute Deviation (MAD), representing the random error. The MAD is preferable to the standard deviation because it is less sensitive to outliers (Ruppert., 2011). We refer to the scaled MAD as random error. The standard deviation is also used in specific cases. The bias, standard deviation, and scaled MAD are calculated as:

$$bias = \sum_{i=1}^{N} HLOS_{observation-instrument}(i)$$

(4)

$$SD = \sqrt{\frac{1}{N-1} \sum_{i=1}^{N} (HLOS_{observation-instrument}(i) - bias)^2}$$

(5)

$$scaled\ MAD = 1.4826 * median\left(\left| HLOS_{observation-instrument}(i) - median(HLOS_{observation-instrument}(i)) \right|\right)$$

(6)

Where 1.4826 represents the scale factor, i the index and N the number of samples. Pearson's correlation coefficient, R, between the HLOS Aeolus wind component and the HLOS instrument component sample is calculated using Eq.7, where $xi$ and $yi$ represent the sample i point of Aeolus and the instrument, respectively. $\bar{x}$ and $\bar{y}$ represent the mean wind component of the datasets.



$$R = \frac{\sum_{i=1}^{N}(xi - \bar{x})(yi - \bar{y})}{\sqrt{\sum_{i=1}^{N}(xi - \bar{x})^2}\sqrt{\sum_{i=1}^{N}(yi - \bar{y})^2}}$$

(7)

The scaled MAD is identical to the standard deviation (Eq. 5) if the analyzed data follows a normal distribution. In addition to the metrics presented above, a least-square line fit to the respective datasets is performed, to also provide the slope and the approximated bias, which we refer to as intercept.

## 3 AboVE campaigns overview

### 3.1 AboVE Maido

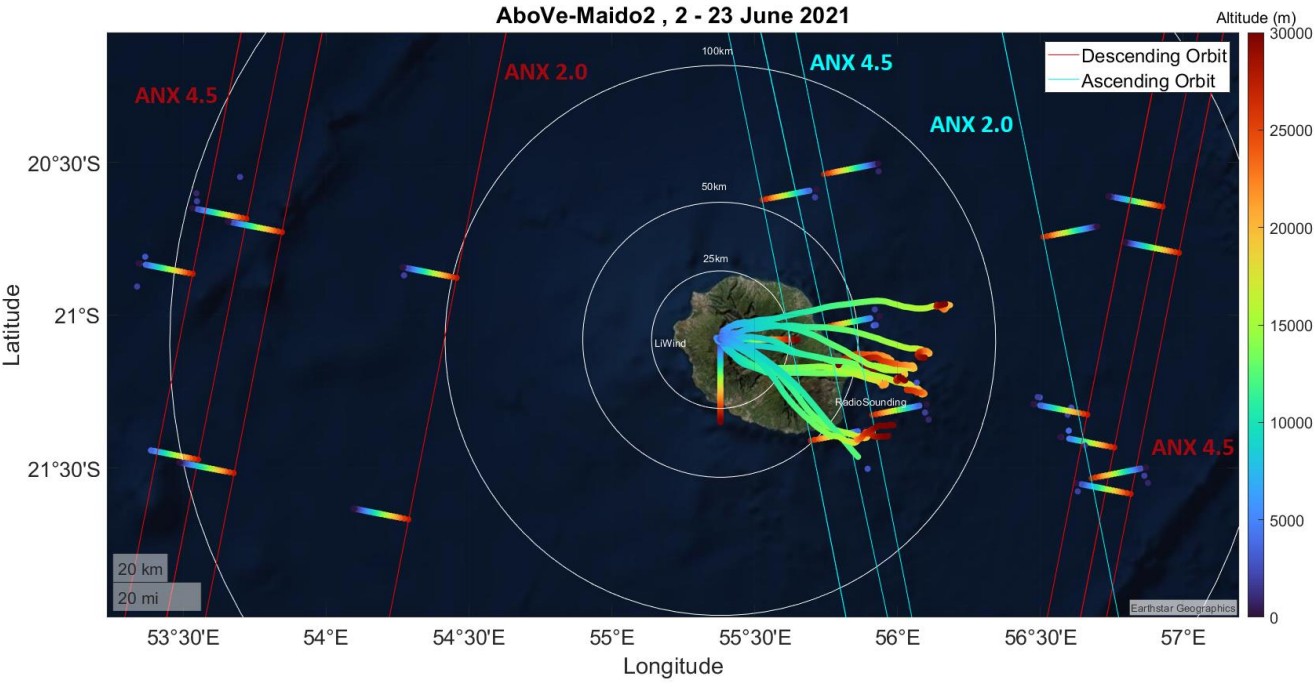

**Figure 1. AboVE-Maido 2 Campaign overview with ascending and descending Aeolus overpasses in cyan and red. The colored trajectories represent the ascending phase of the radiosondes, with their altitude shown through a colormap. The ascending node crossing configuration is specified for each Aeolus overpass. Credit: base map sources are Earthstar Geographics.**

The AboVE-Maido campaigns took place from September 25 to October 10, 2019 (AboVE-Maido1) and from May 31, to June 24, 2021 (AboVE-Maido2). Eleven additional measurements were conducted outside the campaign's dates. The





campaigns took place at the high-altitude Maido Observatory on the French island of La Réunion. Cal/Val activities at the
observatory included Doppler-Rayleigh lidar operation, ranging from early night measurements to dusk till dawn observations
(depending on the overpass's time), to cover both ascending and descending orbits as well as time-coordinated radiosondes
ascents during and in between the overpasses.

During the campaign, the ANX (Ascending Node crossing, the intersect of Aeolus orbit with the x-y plane in Earth fixed
coordinate system) orbit parameter was switched from ANX 4.5 to ANX 2.0 (see Fig. 1 for orbit details). The consequence of
that change was a modification in the orbit's location relative to the observatory. The location of the ANX 4.5 ascending orbit
with respect to the lidar's eastward line-of-sight enabled spatial collocation of the measurements better than 10 km in the lower
stratosphere every Wednesday. The change from the old reference orbit ANX4.5 to the new reference orbit ANX 2.0 occurred
in June 2021. The new orbit was chosen to support the Aeolus tropical campaign activities in Cape Verde and remained. After
the change to the ANX 2.0 configuration, the ascending orbit has moved away from the lidar's eastward line of sight, however
still remained within 100 km.

Thanks to coordination with ESA, the AboVE-Maido 2 campaign took advantage of specific settings planned in advance.
A particularity of this campaign is its unique range bin setting: the Réunion RBS. This specific setting raises Aeolus' top
altitude to 28.7 km which permits intercomparisons higher into the stratosphere. This increase in vertical range is possible at
the expense of the resolution (see Sect 2.3). It is only activated when the satellite overpasses the vicinity of the island. After
the AboVE2-Maido campaign, the cal/val measurement sessions at Maido were conducted once per month until the end of
2021. The OPAR lidar lines of sight facing east, south, and zenith are shown in Fig. 1. The eastward line-of-sight is such that
the lidar acquisition area is crossing with that of Aeolus in the stratosphere (around 40 km altitude), reducing the influence of
the spatial offset. The minimum spatial offset between Aeolus ascending phase and reference measurements was 22.1 km and
22.6 km for the lidar and radiosondes, respectively. At the same time, the descending Aeolus orbits were much further away,
with the spatial offset ranging from 54 km to 241.4 km. During both campaigns, 19 Aeolus-collocated RS ascends were carried
out, of which.15 time-coordinated with ground-based lidar acquisitions. The baseline, date, distance to collocation, bias,
standard deviation, and scaled MAD for both reference instruments, Aeolus overpass time, and orbit type are provided in Table
1.





| Date | RS Distance (km) | RS Bias (ms⁻¹) | RS MAD (ms⁻¹) | Lidar Distance (km) | Lidar Bias (ms⁻¹) | Lidar MAD (ms⁻¹) | Aeolus orbit type |
|---|---|---|---|---|---|---|---|
| 2019-09-25 | x | x | x | 107.7 | 3.06 | 5.62 | Desc |
| 2019-09-25 | 38 | 0.55 | 3.79 | 46.2 | -1.05 | 5.3 | Asc |
| 2019-09-26 | 219.5 | 1.01 | 2.46 | x | x | x | Desc |
| 2019-10-02 | 43 | -2.31 | 1.66 | 29.4 | 0.47 | 4.01 | Asc |
| 2019-10-03 | 201.3 | -1.2 | 3.27 | 206 | -0.27 | 3.44 | Desc |
| 2019-10-09 | 22.7 | -1.1 | 3.69 | 22.6 | -2.93 | 3.67 | Asc |
| 2019-10-10 | 221 | 0.56 | 4.22 | 221.2 | 3.34 | 3.79 | Desc |
| 2019-11-13 | x | x | x | 48.3 | 0.44 | 15.13 | Asc |
| 2020-01-22 | 38.6 | -2.46 | 4.12 | x | x | X | Asc |
| 2020-02-19 | x | x | x | 26.20 | -3.5 | 3.12 | Asc |
| 2020-10-28 | 27.7 | -1.96 | 2.62 | 28.10 | -3.8 | 6.76 | Asc |
| 2021-01-13 | 58.3 | -2.38 | 7.32 | x | x | x | Asc |
| 2021-02-24 | 66.5 | 0.46 | 6.36 | 40.1 | -0.68 | 11.38 | Asc |
| 2021-06-02 | 33.0 | 1.9 | 7.15 | 26.2 | 2.12 | 5.44 | Asc |
| 2021-06-03 | 224.2 | -0.94 | 5.81 | 206.9 | -0.19 | 5.96 | Desc |
| 2021-06-09 | 33.1 | 2.52 | 6.75 | 40.5 | 2.22 | 9.05 | Asc |
| 2021-06-10 | 241.4 | 1.09 | 6.05 | 222.3 | 0.2 | 7.26 | Desc |
| 2021-06-16 | 113.4 | 2.38 | 4.71 | x | x | x | Desc |
| 2021-06-16 | 34.6 | 1.46 | 5.11 | 49.7 | 0.95 | 4.07 | Asc |
| 2021-06-17 | 216.2 | 0.12 | 7.77 | 193.2 | -1.83 | 5.87 | Desc |
| 2021-06-23 | 118.6 | -1.83 | 7.36 | 131.4 | -0.84 | 6.58 | Asc |
| 2021-06-24 | 141.7 | 0.82 | 5.58 | 127.6 | 1.41 | 4.19 | Desc |
| 2021-09-29 | x | x | x | 119.3 | -4.28 | 13.11 | Asc |
| 2021-11-03 | x | x | x | 122.6 | -2.78 | 8.20 | Asc |
| 2021-12-01 | x | x | x | 109.9 | -1.48 | 6.51 | Asc |


**Table 1. Overview of AboVE-Maido 1 and 2 complete list of Aeolus overpasses collocated. The cases of single collocations are also included. The distance is calculated over the average position of each instrument. A more detailed table is provided in the supplement.**





**3.2 Above OHP**

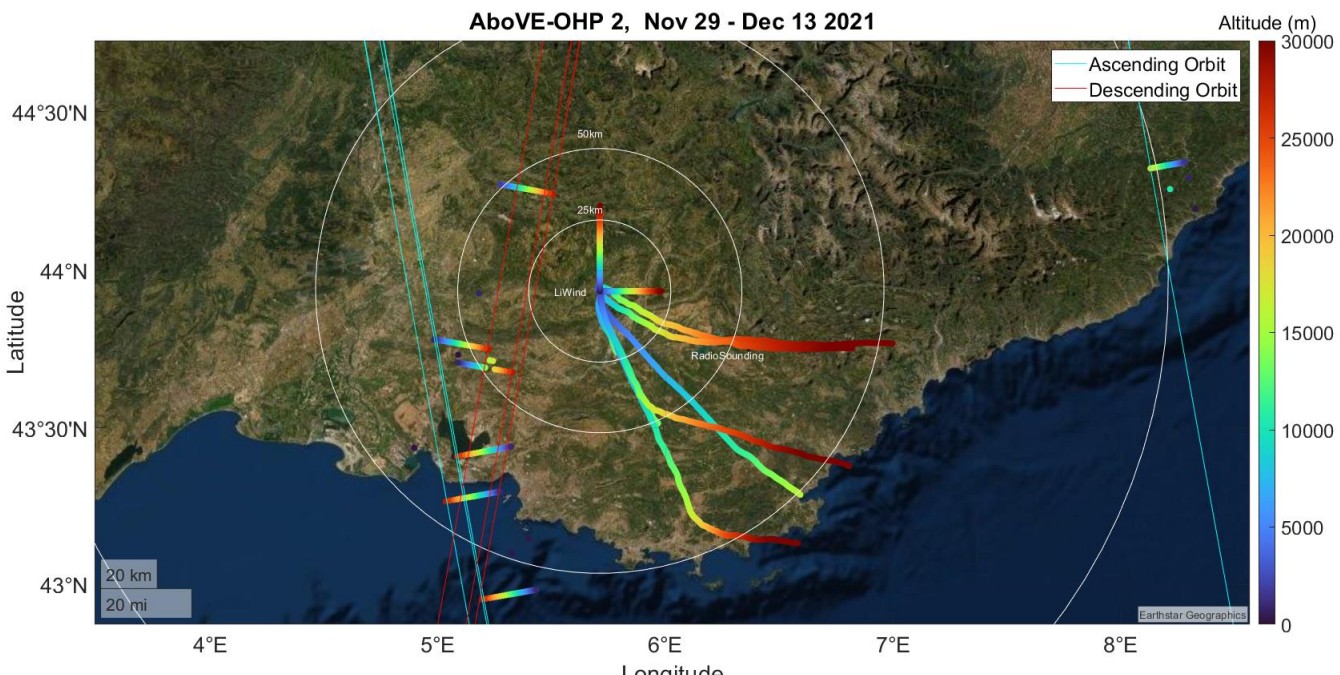

**Figure 2. AboVE-OHP 2 Campaign overview with ascending and descending overpasses in cyan and red. The colored trajectories represent the ascending phase of the radiosondes, with their altitude shown through a colormap. Credit: base map sources are**
**Earthstar Geographics.**

The AboVE-OHP campaigns took place from January 6 to January 14, 2019 (AboVE-OHP1) and from November 29 to

December 14, 2021 (AboVE-OHP2). Additional measurements were conducted on December 20 and 21, 2021, and January

10, 17, and 24, 2022. The ANX 4.5 orbit, active during AboVE-OHP1, allowed for collocations with both ascending and

descending orbits, within 100 to 150 km. The AboVE2-OHP benefited from the satellite orbit modification: The ANX 2.0

orbit enables collocations within 50 km twice per week for ascending and descending orbits. Similar to the AboVE-Maïdo2

campaign, the measurements included dusk till dawn coverage, shorter measurements, and radiosondes ascents. The satellite's

AboVE2-OHP range bin setting is the same as the AboVE-Maïdo2 RBS, since the goal is to compare the two regions' datasets

and differentiate biases between the ones inherent to the setting and the ones due to the geographical location.






| Date | RS Distance (km) | RS Bias (ms$^{-1}$) | RS MAD (ms$^{-1}$) | Lidar Distance (km) | Lidar Bias (ms$^{-1}$) | Lidar MAD (ms$^{-1}$) | Aeolus orbit type |
|---|---|---|---|---|---|---|---|
| 2019-01-06 | 168.0 | 2.52 | 3.72 | 149.7 | 1.04 | 8.31 | Asc |
| 2019-01-07 | x | x | x | 156.9 | 3.22 | 6.74 | Desc |
| 2019-01-07 | 86.2 | -2.47 | 6.55 | 113.1 | -0.14 | 5.32 | Asc |
| 2021-12-06 | 78.3 | -2.70 | 6.12 | 118.3 | -3.24 | 9.29 | Asc |
| 2021-12-07 | 76.2 | -0.20 | 6.12 | 61.3 | -6.81 | 5.01 | Desc |
| 2021-12-12 | 206.8 | -5.05 | 9.55 | 195.2 | -3.62 | 11.83 | Asc |
| 2021-12-13 | 75.0 | -3.35 | 3.82 | 94.6 | -3.11 | 3.26 | Asc |
| 2021-12-14 | 58.5 | 1.50 | 2.85 | 65.5 | 0.47 | 3.22 | Desc |
| 2021-12-20 | 74.5 | -6.56 | 6.42 | 77.9 | -6.53 | 6.59 | Asc |
| 2021-12-21 | 61.8 | -1.08 | 4.15 | 53.9 | -1.34 | 4.56 | Desc |
| 2022-01-10 | 56.4 | -1.23 | 8.16 | 103.4 | 0.06 | 7.28 | Asc |
| 2022-01-17 | 40.4 | -5.36 | 4.27 | 61.8 | -4.50 | 4.35 | Asc |
| 2022-01-24 | 42.4 | 0.69 | 4.56 | 64.1 | 0.61 | 4.45 | Asc |

**Table 2. Overview of AboVE-OHP 1 and 2 complete list of Aeolus overpasses collocated. The distance is calculated over the average position of each instrument. A more detailed table is available in the supplement.**

## 4 Results

### 4.1 Statistical comparison of Aeolus with Collocated Data

In this subsection, we statistically analyze the comparisons, before discussing Aeolus' capacities and performance at different altitudes. The dataset presented consists of the combined measurements from both cal/val sites, including all time periods. For this analysis, we will study the mean bias as a function of altitude, along with the number of collocated data points in Fig. 3. Additionally, we will present the Aeolus Rayleigh wind values plotted in Fig. 4 against the corresponding values of the reference instruments downsampled to match the Aeolus height resolution (as discussed in Sect. 2.3). We provide an overview

of all the validation cases in Tables 1 and 2. Concerning the two reference instruments, a point-by-point comparison shows a mean bias of 0.1 ms$^{-1}$ between the wind profile of the lidar and that of the radiosonde, with a standard deviation of 2.3 ms$^{-1}$.

Within the dataset, distances to collocation vary between 22.1 km and 207 km for ascending orbits and between 54 and 241.4 km for descending orbits. Therefore, the spatial offsets are highly variable, depending on the orbit phase and the ascending node crossing (ANX) setting: For ascending orbits, the average distance to collocation is 67.4 km for radiosondes

and 79.1 km for lidars, whereas, for descending orbits, the averages are 161.4 km and 147.5 km, respectively. Such a difference does not allow for comparing the whole dataset as a unified set of collocations. The reference measurements were separated into ascending and descending orbital phases to account for that disparity.





**Figure 3. a)** The Aeolus minus the radiosonde HLOS wind difference made during all AboVE campaigns over each altitude bin. **b)** The number of data samples over each altitude bin for the radiosonde comparison. **c)** The Aeolus minus the lidar HLOS wind difference made during all the campaigns over each altitude bin. **d)** The number of data samples over each altitude bin for the Lidar comparison. Red represents measurements of an ascending orbit, while black represents measurements of a descending orbit. The lines represent the average bias of each bin altitude, and the shading is the standard deviation of the bias in each range bin.



Figure 3a, 3c, show the mean bias as a function of altitude with the shading representing the standard deviation. From 5 km to approximately 22 km, the bias lies within the +/- 5 ms⁻¹ range but increases as soon as the upper bins mark is reached. The data below 5 km has a larger standard deviation, which is consistent with what Guo et al. (2021) reports on the increased wind speed differences in the 2-3 km range for descending orbits. The same calculations were then realized, only this time within the 5-22 km window (we refer to it as the "Altitude Range Method"). The results show that removing the higher bins

decreases the random error, from 5.58 ms⁻¹ to 5.38 ms⁻¹ for ascending profiles and from 4.99 ms⁻¹ to 4.77 ms⁻¹ for descending profiles. One other method also considered was to average every profile within a window of 200 km around the observatory for each collocation, aggregating 2 or 3 profiles on average (we refer to it as the "Average Method"). With this averaging method, the random error decreases from 5.58 ms⁻¹ to 3.67 ms⁻¹ for ascending profiles and from 4.99 ms⁻¹ to 3.38 ms⁻¹ on descending profiles. The larger standard deviation in the lower troposphere might be due to several reasons. First, the satellite's

lidar performance is largely limited by received power, therefore the strong aerosol scattering in the boundary layer height will lower the apparent molecular scattering signal, reducing the inversion accuracy of HLOS wind from Aeolus (Tan et al., 2017). Secondly, there is also a smaller sample studied in that altitude region, which leads to an undersampling bias. Only Above-OHP2 lidar measurements benefited from an extended coverage below 5 km, which drastically restricts the number of data points.

The same observations still hold true for Fig. 3c, 3d, depicting the same increase in variance in the uppermost bins. Similarly, from 5 to around 22 km, the bias fluctuates within the +/- 5 ms⁻¹ range, and the uppermost bins display similar features in terms of magnitude to the radiosonde's counterpart. The results differ slightly from the radiosondes observations, particularly when applying the methods of Altitude Range or Average on the descending orbits collocations. Indeed, removing the higher bins or averaging them decreases the random error from 7.17 ms⁻¹ to 6.49 ms⁻¹ or 4.9 ms⁻¹ (for removing the higher

and lower bins or averaging several profiles together, respectively) for ascending profiles. This observation also holds for descending profiles, where the random error varies from 7.17 ms⁻¹ to 6.49 ms⁻¹ using the altitude range method and decreases to 3.96 ms⁻¹ with the averaging method. The result of these different methods over the various parameters is reported in table 4. Ground-based lidar comparisons suffer from a higher random error than their radiosondes counterparts. The ground-based lidars show a higher random error in the uppermost bins because its precision decreases with air density (Khaykin et al., 2020).

The lack of data points in the 26 - 27 km range is a specificity of the range bin setting, being the result of a compromise between height coverage and sample spacing.





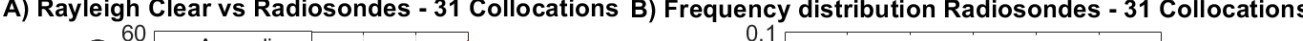

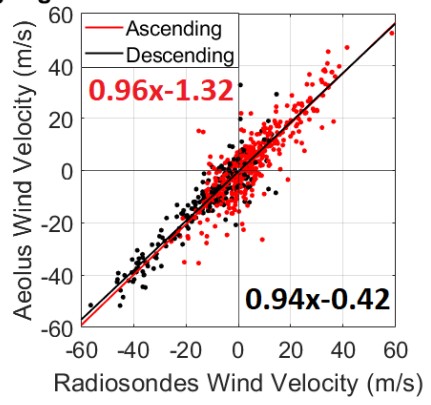
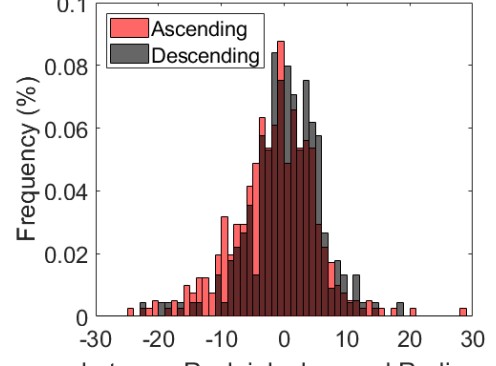

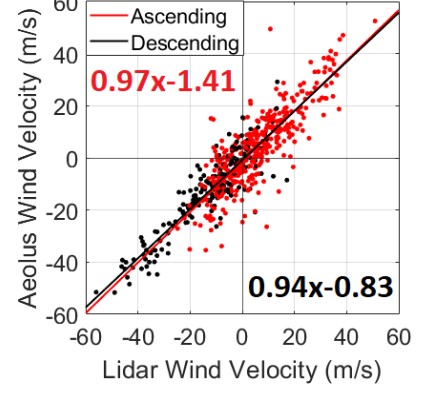
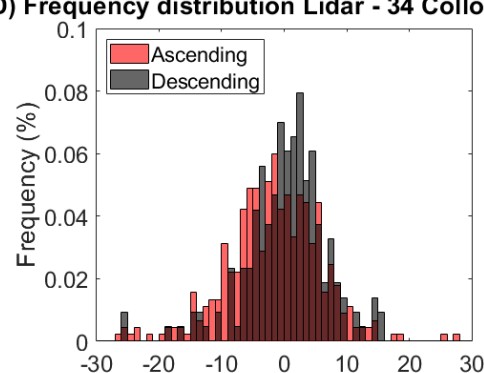

**Figure 4.** **a) The Aeolus winds versus the radiosonde measurements made during all the campaigns. b) Frequency distribution of the difference between Aeolus and radiosonde wind speeds for the same data set. Red represents measurements of an ascending orbit, while black represents measurements of a descending orbit. c) The L2B Rayleigh clear winds versus the lidar measurements made during all the campaigns. d) Frequency distribution of the difference between Aeolus and lidar wind speeds for the same data set. Red represents measurements of an ascending orbit, while black represents measurements of a descending orbit.**

The various colors indicate if Aeolus had an ascending (red) or descending (black) node, i.e., if it was done in the evening or in the morning (local time), respectively. This separation between orbital phases is done because of the aforementioned distance to collocation disparities. Nonetheless, several long-term cal/val activities showed significant phase-dependent differences in the determined biases of Aeolus wind measurements (Wu et al., 2021, Iwai et al., 2021; Lux et al., 2021; Baars et al., 2020; Rennie and Isaksen, 2020; Geiß et al., 2019; Krisch and the Aeolus DISC, 2020), further assessing the need for an independent study on both cases. The correlation plot of the Aeolus wind is shown in Fig. 5a, alongside the retrieved linear regression. A linear trend is clearly seen between the Aeolus and the reference radiosonde observations. The following coefficients are presented with confidence bounds of one sigma. The Fig. 5a ascending trend line has a slope of 0.96 +/- 0.03 with an intercept (i.e., an approximated bias) of -1.32 +/- 0.38 ms$^{-1}$, whereas the descending trend line has a slope of 0.94+/-



0.03 and an intercept of -0.42 +/- 0.56 ms$^{-1}$. The slopes are statistically similar, but the intercepts aren't. Comparing this result
with radiosondes from other cal/val campaigns, Iwai et al. (2021) found a slope of 1.01 and an intercept of 0.38 ms$^{-1}$ in
Okinawa. Baars et al. (2020) presents a slope of 0.97, which is consistent with our observations, and an intercept of 1.57 ms$^{-1}$
that could be explained by the lower maximum height of comparison, where we see the biggest shift towards negative biases
(Fig. 3).

Because of its similar location and window of acquisition, the lidar data share a lot of similar conclusions. In Fig .4c, the
ascending trend line has a slope of 0.97 +/- 0.03 with an intercept of -1.41 +/- 0.45 ms$^{-1}$, whereas the descending trend line
has a slope of 0.94 +/- 0.04 and an intercept of -0.83 +/- 0.78 ms$^{-1}$. The slopes are statistically similar, same as for the intercepts.
Comparing it with other cal/val studies using lidar data, Wu et al. (2021) found overall negative biases with a slope of 1 and
an intercept of -0.12 ms$^{-1}$, confirming the negative intercept we observe for all scenarios. Chen et al. (2022) found slopes of
0.94 and 1.02 and intercepts of -0.89 ms$^{-1}$ and 0.4 ms$^{-1}$ for the lidar large and small geographical range, respectively. Iwai et al.
(2020) found slopes of 0.98 and 1.03 and intercepts of -0.88 ms$^{-1}$ and -0.52 ms$^{-1}$ for the Kobe and Okinawa stations respectively.
Further detail on the results of Aeolus cal/val statistics (including the slope, intercept, bias, scaled MAD, standard deviation,
and correlation coefficient) from various campaigns can be found in Table 6. Figure 4b shows the normalized frequency
distribution of the deviation between the Rayleigh clear and radiosonde wind observations. The distribution follows a Gaussian
pattern, meaning that, according to the normal distribution law, almost 70% of the samples are within the 10 ms$^{-1}$ absolute
error margin. It also seems that the probability of overestimating or underestimating the wind product by Aeolus is
equiprobable. When calculating the mean value of this distribution, one gets -0.79 ms$^{-1}$ as the bias for the Rayleigh wind
observations. If one uses the median of the distribution for the bias calculation, one gets a bias of -0.94 ms$^{-1}$, which is a little
more than the result calculated from the mean. All these results are reported in Table 3. The slope is close to one for both cases,
which means that Aeolus excels at resolving wind speed variations, even if a special sparser RBS is used.

Figure 4.d) depicts the normalized frequency distribution of the deviation between Aeolus and lidar wind observations.
When calculating this distribution's average value for ascending orbit collocations (descending orbit collocations), one gets –
1.41 ms$^{-1}$ (0.11 ms$^{-1}$) as the bias for the Rayleigh clear wind observations. If we use the median of the distribution for the bias
calculation, one gets a bias of -1.05 ms$^{-1}$ (0.16 ms$^{-1}$). Putting these two together, the mean value becomes -0.92 ms$^{-1}$ and the
median -0.48 ms$^{-1}$. While close, there is a clear difference in biases based on whether it is using an ascending or descending
orbit in reference, further assessing the need to distinguish both cases.



|  | RS | | Lidar | |
|---|---|---|---|---|
|  | Asc | Desc | Asc | Desc |
| Mean | -1.43 | -0.37 | -1.41 | 0.11 |
| Median | -1.9 | 0.56 | -1.05 | 0.16 |
| Global Mean | -0.79 | | -0.92 | |
| Global Median | -0.94 | | -0.48 | |
| Number of collocations | 20 | 11 | 23 | 11 |

**Table 3. Overview of the different statistics regarding the biases in Aeolus with respect to the reference instruments. The data used includes all the collocations listed in Tables 1 and 2.**

Overall, there is not a better slope or a lower bias for ascending orbits, which would go with the conclusion of the absence of any representative statistical difference presented by Guo et al. (2021). The data even suggests that the bias is lower for descending orbits, which would contradict the fact that they have a larger spatial offset from the reference measurement locations. We can also observe a negative bias for all the comparisons using the standard all-data method. This means that the satellite is prone to underestimating the HLOS wind speeds. A previous study indicates differences in the biases between the ascending and descending orbit phases, mainly occurring for the Rayleigh channel in late summer and autumn (Martin et al., 2021). One reason Sun et al. (2014) raised may be that the meteorological conditions such as wind speeds, Boundary-Layer Height, air temperature, and aerosol distributions differ from one orbital phase to the other. However, none of these orbit-dependent events were observed during our measurements. Other studies indicate lidar comparison biases of 1.05 ms$^{-1}$ and -0.35 ms$^{-1}$ for Chen et al. (2022), -0.13 ms$^{-1}$ for Wu et al. (2021), and -0.81 ms$^{-1}$ -0.48 ms$^{-1}$ in Kobe and Okinawa respectively for Iwai et al. (2021). For radiosonde comparison biases, we observe 2.12 ms$^{-1}$ from Martin et al. (2021), 1.52 ms$^{-1}$ from Baars et al. (2020), and 0.45 ms$^{-1}$ from Iwai et al. (2021). While the lidar comparisons biases are in good agreement with what we observe, the radiosonde biases show a greater variety in the outcomes, which might be due to the pendulum-line motion of the suspended radiosonde during the early stage of ascent (Kumer et al., 2014).





| | | | RS | | Lidar | |
|---|---|---|---|---|---|---|
| | | | Asc | Desc | Asc | Desc |
| All data | | Slope | 0.96 | 0.94 | 0.97 | 0.94 |
| | | Y intercept | -1.32 | -0.42 | -1.41 | -0.83 |
| | | corr coeff | 0.86 | 0.92 | 0.85 | 0.92 |
| | | Scaled MAD | 5.58 | 4.99 | 7.17 | 5.06 |
| Altitude Range | | Slope | 1 | 0.97 | 1.03 | 1 |
| | | Y intercept | -1.12 | 0.45 | -1.8 | 1.17 |
| | | corr coeff | 0.91 | 0.95 | 0.89 | 0.94 |
| | | Scaled MAD | 5.38 | 4.77 | 6.49 | 5.01 |
| Average within 200km | | Slope | 0.98 | 0.92 | 0.99 | 0.93 |
| | | Y intercept | -1.02 | -1.35 | -1.43 | -1.61 |
| | | corr coeff | 0.90 | 0.94 | 0.92 | 0.94 |
| | | Scaled MAD | 3.67 | 3.38 | 4.9 | 3.96 |

**Table 4. Overview of the different comparison method and their relative outcome on the statistical metrics, depending on the instrument and the orbit phase. The data used includes all the collocations shown in Tables 1 and 2.**

Table 4 shows the previously mentioned data and the different methods for comparison. The all-data method means that the instrument is collocated within the closest overall Aeolus overpass. The altitude range method cuts the data outside the 5 to 22 km range to account for possible RBS issues and non-optimal satellite coverage area. The last method aggregates every eligible profile (within the last two hours) into one, averaged, single collocation. The goal was to see if the uppermost bins issue would become insignificant after a certain amount of averaging, hence making the 22+ km range data available for usage. So far, the method shows encouraging data, reducing the scaled MAD for both instruments at any orbit type. Putting these numbers into perspective, for the radiosondes, a scaled MAD of 4.84 ms[-1] is found by Baars et al. (2020), 3.97 ms[-1] for Iwai et al. (2021), and 5.01 ms[-1] for Martin et al. (2021). A standard deviation of 4.43 ms[-1] (Iwai et al. 2021) is also reported. We report a scaled MAD of 5.37 ms[-1] and a SD of 6.18 ms[-1], which belongs to the same range of results, if we account for the fact that we consider a broader altitude range for comparison (see Table. 5).

For the lidars, other cal/val campaigns report scaled MADs of 3.91 ms[-1] (Wu et al. 2022), 5.21 ms[-1], and 5.58 ms[-1] for Iwai et al. (2021). They also report SDs of 5.98 ms[-1] and 4.78 ms[-1] (Chen et al., 2022), 4.76 ms[-1] (Wu et al., 2021), and 5.69 ms[-1]





and 6.53 ms[-1] (Iwai et al., 2021). Our results point to a scaled MAD of 6.49 ms[-1] and a SD of 7.25 ms[-1], further confirming
other studies even if we observe bigger numbers, because of a wider altitude range comparison. It should also be noted that
we do not meet the mission requirements (Ingmann and Straume, 2016), as seen in previous studies (Baars et al., 2020; Iwai
et al., 2021; Martin et al., 2021; Rennie et al., 2021; Wu et al., 2022) with similar magnitudes and values. The offset between
the observed SD and the mission requirement's SD does not seem to change between the free troposphere and the first
kilometers of the stratosphere. The uppermost region of observation is where we reach the highest difference between the
observed and the required SD, by more than 4 ms[-1].

|  | Mission Requirement limits (ms[-1]) | Radiosonde (ms[-1]) | Lidar (ms[-1]) |
|---|---|---|---|
| Bias | 0.7 | -0.79 | -0.92 |
| SD 2-16 km | 2.5 | 5.04 | 5.69 |
| SD 16-20 km | 3 | 4.93 | 5.59 |
| SD 20-30 km | 5 | 9.19 | 9.41 |

**Table 5. Aeolus Mission Requirements on the bias and standard deviation (MR-100 and MR-110 Ingmann and Straume, 2016) compared to the observed bias and standard deviation, depending on the altitude level.**

### 4.2 Case-based analysis

In this subsection, we further discuss Aeolus' capacities and performance with the help of four specific case studies. The goal
is to provide an overview of the measurements by choosing representative collocations, each helping to show specificities
present in the dataset. It shows that in the same conditions (i.e. spatial/temporal offsets) Aeolus can behave very differently.
These measurements are part of a more extensive statistical analysis presented in the above subsection. The figure 5 shows the
HLOS wind velocity profiles measured by the radiosonde (black), the ground-based lidar (red), and the space lidar (blue). The
shadings represent the measurement error for each data point, and the radiosondes' uncertainty is between 0.4 and 1 ms[-1]
(Dirksen et al., 2014), which is smaller than the ground-based lidars (2.2 ms[-1]) (Khaykin et al., 2020) and Aeolus (4.1–
4.4 m s−1) (Martin et al., 2021) uncertainties. For clarity, radiosonde wind speed uncertainties are not plotted in Fig. 5. Since
Aeolus is taking measurements 35° off-nadir, the horizontal distance of the Aeolus observations to the ground-based lidar is
different for each height bin in the Aeolus wind profile. At the same time, the radiosonde drifts along the direction of local
wind, making the distance between the Aeolus measurements and the radiosonde vary during the balloon sounding as a function
of height.





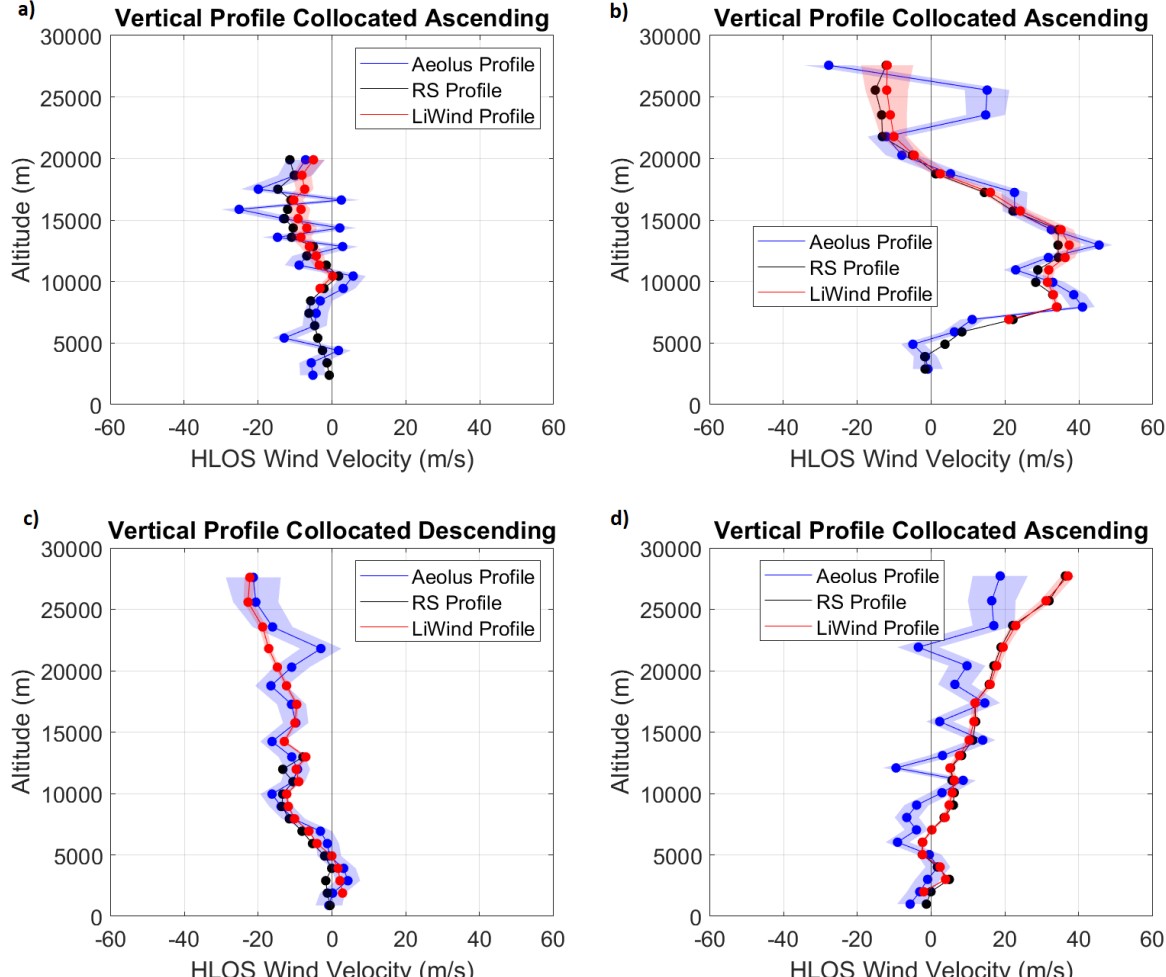

**Figure 5. Wind velocity profiles measured by the radiosonde (black) with the closest Aeolus Level 2B Rayleigh clear (blue) and**
**ground-based lidar (red) profiles. The lidar and radiosonde profiles are shown with an adjusted resolution to the Aeolus range bin**
**width. The lidar and radiosonde measurements are projected to the HLOS of Aeolus. a) February 24, 2021 (Above Maido 2), b) June**
**9, 2021 (above Maido2), c) 14 December 2021 (Above OHP2), d) December 20, 2021 (Above OHP 2). The shading represents the**
**internal estimated error for each instrument.**

Case Study A: February 24, 2021 (Above Maido 2) - 2B11

This collocation was taken at 14h32 UTC. Both instruments were started 30 minutes prior, at around 14h UTC. The mean measuring distance is 67 km for the radiosonde and 40 km for the lidar. The mean bias of the radiosonde is 0.46 ms⁻¹, and the lidars' is -0.68 ms⁻¹. The standard deviation is 6.8 ms⁻¹ and 8.8 ms⁻¹, the scaled MAD is 6.36 ms⁻¹ and 11.38 ms⁻¹, and correlation coefficients are 0.55 and 0.46. The figure depicts a very specific pattern of oscillating nature. Although showing a similar trend to the other two instruments (both mean bias lower than |1| ms-1 for both reference instruments), the ALADIN shows a very



particular signature between 12 and 19 km. This observation could hint toward the existence of instrument-induced oscillating perturbations, I2OPs. We could not find any literature assessing the existence of similar phenomena. Henceforth, what remains to be explained is the nature of the oscillation, which does not correspond to any known phenomenon. While the hot pixels could also play a part in this phenomenon, explaining the increase in magnitude, they cannot account for the oscillating nature (Weiler et al., 2021).

Case Study b):  June 9, 2021 (Above Maido 2) - 2B12

This collocation was taken at 14h33 UTC. Both instruments were started 30 minutes prior, at around 14h UTC. The mean spatial offset is 33 km for the radiosonde and 40.5 km for the lidar. The mean bias between Aeolus and the radiosonde is 2.52 ms$^{-1}$, and the mean bias between Aeolus and the lidars is 2.22 ms$^{-1}$. The standard deviations are 11.21 ms$^{-1}$ and 11.54 ms$^{-1}$, the scaled MAD are 6.75 ms$^{-1}$ and 9.05 ms$^{-1}$, and correlation coefficients are 0.83 and 0.84.

It is found that the Aeolus wind profile in the atmospheric boundary layer and the lower troposphere is in good agreement except for the 22 km height and higher bins of the Aeolus wind profile, which have a significant bias compared with the radiosondes and lidar-retrieved HLOS wind. These exceptionally strong deviations are observed for most AboVE-Maido 2 campaign collocations and happen specifically within the uppermost bins. The low molecular density could explain the cause of the higher values observed at high altitude levels. We conclude that, because of this mission's specific RBS, the satellite is

not resolving the higher altitude ranges with enough precision since it is not receiving enough backscattering.

Case Study c): 14 December 2021 (Above OHP2) - 2B13

The third case study is from the second and most recent cal/val campaign conducted at the Observatoire de Haute Provence. The distance to collocation is under 70 km for both instruments, and the time interval is less than 2 hours. The mean bias between Aeolus and the radiosonde is 1.5 ms$^{-1}$, and the mean bias between Aeolus and the lidars is 0.5 ms$^{-1}$. The standard

deviations are 2.84 ms$^{-1}$ and 4.02 ms$^{-1}$, the scaled MAD are 2.85 ms$^{-1}$ and 3.22 ms$^{-1}$, and correlation coefficients are 0.92 for radiosonde and 0.86 for lidar. The scaled MAD is exceptionally low compared to other cases, showing that ALADIN is still able to perform very well and deliver excellent results, beyond the nominal lifetime of three years. Here, the Rayleigh clear profile obtained by Aeolus is within a close interval to the ground-based data, except for a sudden spike appearing at around 23km. Indeed, this spike not observed on the reference lidar could be linked to Aeolus's hot pixels issue (Weiler et al., 2021a)

since 24 hot pixels were present in the Rayleigh channel (ESA., 2021).

Case Study d): December 20, 2021 (Above OHP 2) - 2B13

This collocation is a good example of the poor-behaving measurements Aeolus can come across because the satellite seems to miss what the lidar and the radiosonde see even within close collocation criteria (40 minutes difference and both instrument under 80 km). An oscillating perturbation is also present on Aeolus' profile as opposed to the reference data. Given the very

close collocation, one could assume a negligibly small geophysical bias that the three observations would keep a general common trend.  We are looking at an oscillating profile with significant offsets from the reference measurement. That could be interpreted as a coupling between both oscillations and hot pixels, meaning that these phenomena can co-occur. The mean bias between Aeolus and the radiosonde is -6.56 ms$^{-1}$, and the mean bias between Aeolus and the lidars is -6.53 ms$^{-1}$. The



standard deviations are 6.84 ms$^{-1}$ and 7.12 ms$^{-1}$, the scaled MAD are 6.42 ms$^{-1}$ and 6.59 ms$^{-1}$, and correlation coefficients are
0.76 and 0.75.

**4.3 Long-term validation**

This section aims to convey an idea of the evolution of Aeolus' performance over its mission cycle, starting in September
2018. For comparisons, Météo-France sites in La Réunion (21° S, 55° E) and Nîmes were used (43° N, 4° E). These sites
perform twice-a-day radiosonde launches at midnight and noon, opening broader possibilities for potential collocations. Both
sites have an average of 120 km of distance to collocation. The La Reunion site has a time difference of 3 hours on average,
whereas the Nîmes site has 5 hours 30 minutes time difference.

Figure 6 shows the time series of standard deviation of the difference between Aeolus Rayleigh clear HLOS wind and
Météo-France radiosonde launches at both sites. Outlier detection was conducted on both time series using the scaled MAD
technique (outliers are defined as elements more than three scaled MAD from the median). Also shown in Fig. 6 are the Cal/Val
campaign observations. For this comparison, we used only Aeolus data between 5 and 22 km to exclude the uppermost bins
characterized by the largest errors (Sect. 4.2).

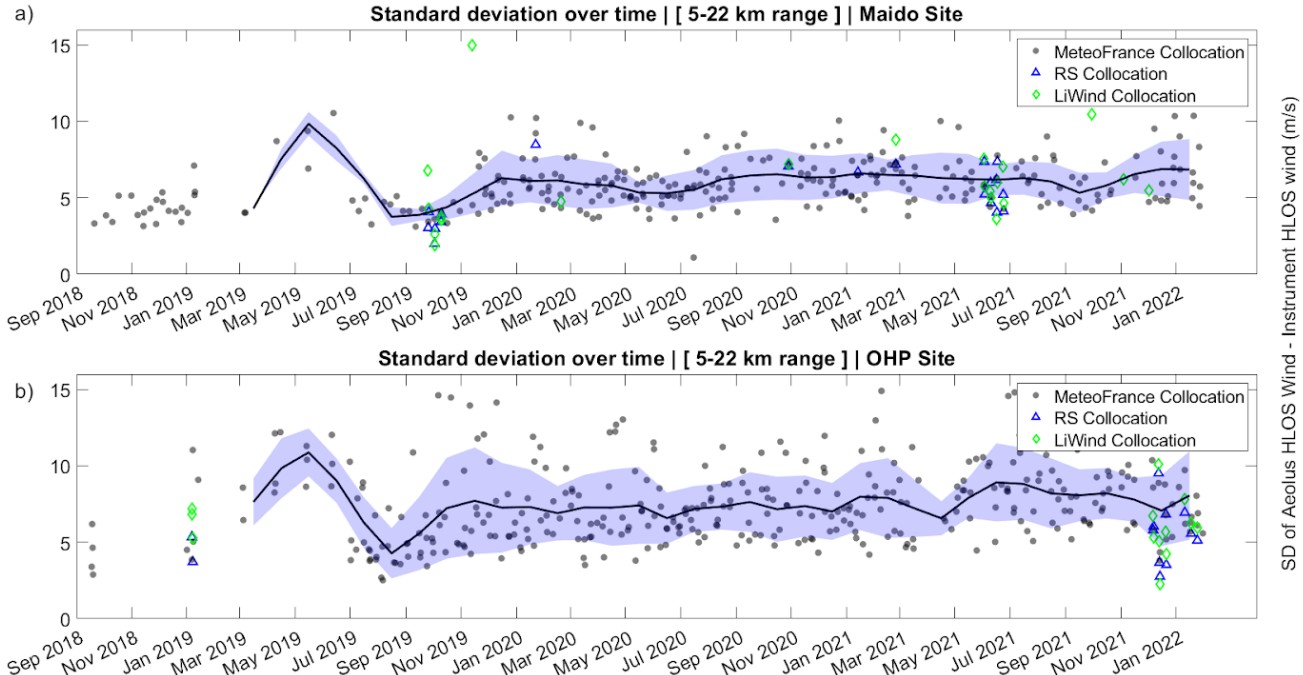

**Figure 6. Standard deviation of the difference between Aeolus Rayleigh clear HLOS wind and Météo-France Radiosonde HLOS
wind, over time a) at the Maido Observatory b) at the Haute Provence Observatory. The Météo-France data were downsampled to
Aeolus vertical resolution and projected to the HLOS of Aeolus. For radiosonde and ground-based lidar collocations, the cal/val
punctual collocations are reported in the figure in blue and green markings.**



As previously mentioned, both sites have a similar average distance to the collocation, however the time offset is larger for Nimes site by 2h30. As demonstrated in Sect.4.1, the distance to collocation does not significantly impact the bias of the collocations. As such, this explains why both time series share a very similar trend and show the same variations specific to mission events (FM-A laser power loss, FM-B switch etc.). We note that the standard deviation at Nimes is higher by 1.5 ms[-1] on average compared to that at La Reunion. Because the distance to collocation is the same at both sites, this higher random error can be explained by the larger time offset. Once can thus conclude that in terms of the collocation criteria, the temporal offset is more critical than the spatial offset when collocating satellite and ground-based wind measurements.

For the La Reunion time series (Fig. 6a), the October 2019 and June 2021 AboVE-Maido campaign data correspond well with the Météo France observations, whereas the lower values could be explained by the campaign collocations with reduced geophysical bias. Additional lidar observations at Maido conducted outside the dedicated campaigns, show increased random errors, supposedly due to the larger time offsets for some of these collocations (cf. Table 1). The results of the AboVE-OHP-2 campaign, displayed in Fig. 6b, show random errors similar or lower than those inferred from the Météo-France radiosonde launches. As for the La Reunion series, the lower random errors can be explained by the reduced spatial and temporal offsets for the campaign collocations.

Overall, a few conclusions can be shared across both sites of observation. Over time, we observe a slight increase in the standard deviation, (+1 ms[-1] on average between January 2020 and 2022). It is possible to split the time series into different periods. The early FM-B period (July 2019 to January 2020) shows the lowest random error and is consistent across both observation sites. We refer to this period as the "golden era" of Aeolus. Around mid-April 2020 the M1 bias correction introduced into the retrieval, has lowered the bias significantly (Weiler et al. 2021b) however it does not appear to have an impact on the random error. From this point on, the disparity of the data stabilized, as well as the average value. The last months of the time series show a stable average value as well as its standard deviation, asserting that the satellite is maintaining a good precision over the last few months of 2022.

High standard deviation values could be due to hot pixels, and I2OPs mentioned above present in the ACCDs. Previous research revealed that 6% of the ACCD pixels are hot pixels and that around 13% of the pixels will be affected at the end of the extended mission lifetime in November 2022, assuming that the hot pixel generation rate does not change (Weiler et al., 2021a). Corrections to several substantial bias sources in the Aeolus L2B winds have been implemented, including corrections to the dark current signal anomalies of single pixels (so-called hot pixels) on the Accumulation-Charge-Coupled Devices (ACCDs), linear drift in the illumination of the Rayleigh/Mie spectrometers, and the telescope M1 mirror temperature variations (Reitebuch et al., 2020; Weiler et al., 2021b). Indeed, figure 6 confirms other studies that claim that hot pixels, laser energy, and receiving path degradation effects in ALADIN have been mitigated (Feofilov et al., 2022; Baars et al., 2020; Weiler et al., 2020). If this mitigation were not present, we would observe an increase in the standard deviation at a much higher rate (Weiler et al., 2021a). Unfortunately, due to potential calibration issues, uncorrected biases might remain in Aeolus L2B winds and may contribute to potential biases between Aeolus and the Météo-France observed winds. In addition, the Aeolus L2B winds might be biased towards the ECMWF model, as the M1 bias correction makes use of ECMWF 6-hour



forecasts (Rennie et al., 2021), which might also lead to suboptimal assimilation of Aeolus winds on ground comparisons (Liu et al., 2022).

## 5. Discussion

In this paper, to evaluate the accuracy and precision of the Aeolus retrieved wind results, double collocation techniques using both radiosondes and ground-based lidars were conducted. Due to the proximity of collocation and a similar functioning principle, a statistical analysis of random error, biases, and their evolution throughout the mission lifetime was performed, from the observation sites of Haute-Provence and La Réunion. A total of 65 collocations were collected during periods from baseline 2B02 to 2B13.

| Rayleigh-clear | | | | | | | |
|---|---|---|---|---|---|---|---|
| Campaigns and instruments | | R | SD | Scaled MAD | Bias | Slope | Intercept |
| | | | $ms^{-1}$ | $ms^{-1}$ | $ms^{-1}$ | | $ms^{-1}$ |
| Radiosondes (This study) | | 0.89 | 6.18 | 5.37 | -0.79 | 0.95 | -0.87 |
| Ground Lidars (This study) | | 0.77 | 7.25 | 6.49 | -0.82 | 0.96 | -1.12 |
| AVATAR-T (Witschas et al., 2022) | | 0.52 | 7.7 | 7.1 | -0.1 | 0.97 | -0.2 |
| AVATAR-I (Witschas et al., 2022) | | 0.91 | 5.6 | 5.5 | -0.8 | 1 | -0.8 |
| WPR over Australia (Zuo et al., 2022) | | 0.92 | 6.22 | 5.81 | -0.48 | 1.02 | -0.44 |
| USTC - LGR (Chen et al., 2022) | | 0.96 | 5.98 | – | 1.05 | 0.94 | -0.89 |
| USTC - SGR (Chen et al., 2022) | | 0.98 | 4.78 | – | -0.35 | 1.02 | 0.4 |
| Radiosondes (Martin et al., 2021) | | – | – | 5.01 | 2.12 | – | – |
| VAL-OUC (Wu et al., 2022) | 2B07/08 | 0.39 | 10.2 | 8.42 | -1.23 | 1.12 | -1.16 |
| | 2B09/10 | 0.75 | 4.66 | 3.84 | -0.98 | 0.97 | -1.01 |
| | 2B11 | 0.86 | 4.76 | 3.91 | -0.13 | 1 | -0.12 |
| WindVal III /A2D improved (Lux et al., 2022) | | – | 7.4 | 7.4 | -0.85 | – | – |
| WindVal III/A2D (Lux et al., 2020a) | | 0.8 | 3.6 | 3.6 | 2.6 | – | – |
| WindVal III/2 µm DWL (Witschas et al., 2020) | | 0.95 | 4.75 | 3.97 | 2.11 | 0.99 | 2.23 |
| AVATARE (Witschas et al., 2020) | | 0.76 | 5.27 | 4.36 | −4.58 | 0.98 | −4.39 |
| AboVE-OHP (Khaykin et al., 2020) | | 0.96 | 3.2 | – | 1.5 | – | – |
| RV Polarstern cruise PS116 (Baars et al., 2020) | | – | – | 4.84 | 1.52 | 0.97 | 1.57 |
| MARA (Belova et al., 2021) | summer | 0.82 | 5.8 | – | 0 | 1.1 | 0 |





| | | | | | | | |
|---|---|---|---|---|---|---|---|
| | winter | 0.81 | 5.6 | – | −1.3 | 0.87 | −0.8 |
| ESRAD (Belova et al., 2021) | summer | 0.92 | 4.5 | – | −0.4 | 1 | −0.5 |
| | winter | 0.88 | 5.2 | – | −0.4 | 1 | −0.6 |
| WPR over Japan (Iwai et al., 2021) | 2B02 | 0.95 | 8.08 | 7.35 | 1.69 | 0.98 | 1.75 |
| | 2B10 | 0.9 | 7.89 | 7.08 | −0.82 | 0.94 | −0.74 |
| CDWL in Kobe (Iwai et al., 2021) | 2B02 | 0.98 | 6.17 | 4.92 | 0.46 | 1.05 | 0.61 |
| | 2B10 | 0.96 | 5.69 | 5.21 | −0.81 | 0.98 | −0.88 |
| CDWL in Okinawa (Iwai et al., 2021) | 2B02 | 0.93 | 6.57 | 5.68 | 1.08 | 0.99 | 1.07 |
| | 2B10 | 0.79 | 6.53 | 5.58 | −0.48 | 1.03 | −0.52 |
| GPS-RS in Okinawa (Iwai et al., 2021) | 2B02 | 0.99 | 4.55 | 4.77 | 1 | 0.99 | 1 |
| | 2B10 | 0.99 | 4.43 | 3.97 | 0.45 | 1.01 | 0.38 |
| RWP network over China (Guo et al., 2021) | | 0.81 | 6.82 | – | -0.64 | 0.99 | -0.64 |

**Table 6. Summary of the recent comparison campaign validation results, following Wu's (2021) referencing convention**

Similar to Wu's (2021) initiative, we summarized the recent comparison campaigns from the Cal/Val teams across the globe and Aeolus's lifespan. We decided to use the same statistical parameters and presentation that Wu (2021) proposed, in order to keep a consistent comparison method over several studies. Therefore, the statistical parameters include the correlation coefficient, SD, MAD, bias, slope, and intercept. We can compare our results to other instruments that weren't mentioned

during previous sections, such as ALADIN Airborne Demonstrators (A2D) (Witschas et al., 2020; Lux et al., 2020a; Lux et al., 2022), airborne Doppler Wind Lidars (DWL) (Witschas et al., 2020; Witschas et al., 2022) and Radar Wind Profilers (WPR) (Zuo et al., 2022; Guo et al., 2021; Belova et al., 2021; Iwai et al., 2021). Close to Wu's (2021) observations, we observe the same consistency and similarities with the more recent studies. A non-negligible proportion of the disparities between the measurements are caused by the variety of comparison ranges, ranging from the boundary layer to the mid-

stratosphere.

Over time, we assess a very modest decrease in the precision of ALADIN measurements (Fig. 6), at both cal/val sites. While studies claim that the effects have been mitigated thanks to various corrections (Rennie et al., 2021 Reitebuch et al., 2020; Weiler et al., 2021b), it might not compensate for the increased error over time, coming from hot pixels, the temperature of the mirror and laser-induced contamination. Uncorrected biases might remain in Aeolus L2B winds and may contribute to

potential biases between Aeolus and the Météo-France observed winds (Liu et al., 2022).

The argument made in Sect.4.3 is that the time offset is more important than the spatial offset, in terms of impact on the collocation quality. Therefore, it should be inferred from this result that the time criterion has to be favored over the distance criterion when collocating different measurements. To do so, we showed how the distance to collocation had small effects on the bias in Sect.4.2 and how the time difference had a significant impact on the standard deviation in Sect.4.3. Some error

sources were pointed out by previous research, e.g., hot pixels and dark current anomalies (Weiler et al., 2021a), Rayleigh



wind errors introduced by angular variations (Lux et al., 2022, Lux et al., 2018, Lux et al., 2020a), vibrations introduced by the satellite platform, which affects the Q-switched master oscillator cavity length (Lux et al., 2020b), photon shot noise (Liu et al., 2006), micro-vibrations due to critical rotation speeds of the satellite's reaction wheels (Lux et al., 2021), mechanical disturbances generated by reaction wheels of the class of those embarked on Aeolus (Le., 2017), linear drift in the illumination

of the Rayleigh/Mie spectrometers, and the telescope M1 mirror temperature variations (Reitebuch et al., 2020; Weiler et al., 2021b). In section 4.1, we observe a steep increase in the random error above 25 km, which can be further observed in fig.5b. We believe that this issue, specific to the high RBS profiles, is caused by the lack of molecular backscatter. The lack of signal could result in a higher random error, but the estimated error is shown to still be within a narrow interval (+/-5 ms$^{-1}$), compared to its observed deviation from reference measurements. Although bigger than what is shown for the previous altitude bins, the

retrieved measurement error for the uppermost bins is strongly underestimated, along with the physical fact that there are fewer particles at higher altitude levels.

In this study, we also refer to I2OPs, which might be a new, unreferenced, phenomenon. Over the 65 collocations, the phenomenon was observed 5 times. It is impossible to tell how recurrent this perturbation appears, because it requires a single profile-to-profile reference measurement, which is not something done in Aeolus to models comparisons. We note that

applying a high-pass filter with the cutoff at 5 km vertical wavelength smooths out all oscillations and provides an estimation of the profile much closer to that of the reference instruments. While this is a good patch, it is not possible to apply such a filter on every wind profile, as it would probably also remove traces of physical events. One way to further improve this method, would be to see how the HLOS value fluctuates from one bin to another. Since I2OPs tend to have a fairly constant peak-to-peak amplitude and period it should be fairly easy to detect such patterns in the data. One way to further track down

this issue would be to retrieve the L1B useful signal, as well as the Signal to Noise Ratio (SNR), to see if it could be linked to bad behavior of the optical system itself.

We observe a negative bias in most scenarios, except for the Altitude range method, where the descending orbit lidar collocations show a positive bias. On average, the satellite tends to underestimate the wind speed by around 1 ms$^{-1}$. Several papers cited above (Sect.4.1 and Table 6) report similar observations.

**6. Summary**

The Aeolus wind products from the first wind lidar in space, ALADIN, were validated against radiosondes and ground-based Doppler Rayleigh-Mie Lidars from the observation sites of OPAR in La Réunion and OHP in Haute Provence (France). All 65 collocations were collected during periods from baseline 2B02 to 2B13, spanning January 2019 to January 2022. In summary, we find a standard deviation of 6.18 ms$^{-1}$ and 7.25 ms$^{-1}$ and a scaled MAD of 5.37 ms$^{-1}$ and 6.49 ms$^{-1}$ for radiosondes

and lidars intercomparisons, respectively. We also find correlation coefficients of 0.82 and 0.77, slopes of 0.95 and 0.96 and y-intercepts of -0.87 ms-1 and -1.12 ms-1 for radiosondes and lidars intercomparisons, respectively. The biases and random errors observed are higher than those outlined in the mission requirements (Ingmann and Straume, 2016), as seen in Table 5:





The bias is higher by and average of 0.15 ms-1 and the standard deviation was exceeded at every altitude level by 50%, on average.

It was shown in Sect.4.3 that in terms of collocating the ground-based and satellite measurements, the time criterion would be more prevalent than the distance criterion when choosing which one to favor. This study also showed previously undocumented phenomena (Instrument Induced Oscillating Perturbations, I2OPs Sect.4.2, excessive random error in the uppermost bins above 25 km, Sect.4.1. While the latter phenomenon can be mitigated through the use of averages between several adjacent profiles, the I2OP issue could be addressed by application of frequency filters, which would require further

investigation.

        Within this study, we have noticed range-bin and temporal wind dependencies. For the uppermost bins (above 22 km on average) enabled by the AboVE RBS, the random error is enhanced by 2 - 3 ms-1 for ascending and descending phases. This can be explained by the lower air density there, reducing the molecular backscatter intensity. We note that aggregation of two or more adjacent Aeolus profiles improves the comparison by 70%. For the larger spatial offsets, the method yields poorer

results compared to the altitude range method. Both methods lowered the scaled MAD on any comparison category for any instrument. Simlarly to the results by Guo et al. (2021) and Zuo et al. (2022), we do not observe any significant difference between ascending and descending phases which goes against previously observations about orbit-dependent characteristics (Rennie et al., 2021, Martin et al., 2021). Guo et al. (2021) showed slopes of 0.91 and 0.96 and intercepts of 0.47 ms-1 and -1.4 ms-1, respectively. This is close to our observations. Detailing our results, the comparison of ascending phase for radiosondes

collocations has a mean correlation coefficient of 0.77 and a scaled MAD of 5.58 ms-1. In contrast, the descending phase has a mean correlation coefficient of 0.91 and a scaled MAD of 4.99 ms-1. The lidars collocations in ascending phase have a mean correlation coefficient of 0.73 and a scaled MAD of 7.17 ms-1, whereas the descending phase has a mean correlation coefficient of 0.85 and a scaled MAD of 5.06 ms-1.

        Overall, we recognize evolutions in the L2B data quality throughout the satellite's lifetime. Thanks to the regular

measurement campaigns and the use of twice-daily Météo-France radiosondes, we managed to observe the long-term evolution of the precision of the satellite based on the standard deviation of daily collocations made between the satellite and the closest Météo-France station. As observed during the AboVE-Maido validation campaigns, the mean random error increases from 4.6 ms-1 to 7.6 ms-1 between AboVE-1 (October 2019) and the AboVE-2 campaign (June 2021). For the AboVE-OHP campaigns, the mean random error increases from 5.6 ms-1 to 6.4 ms-1 between AboVE-1 (January 2019) and the AboVE-2 campaign

(December 2021). This is consistent with the routine observations using Météo-France's radiosondes. The early FM-B period, "the golden era" of Aeolus, shows the lowest random error at both sites, which has been increasing ever since.

        *Data availability.* Aeolus Data is publicly available through the Aeolus online dissemination system (https://aeolus-ds.eo.esa.int/oads/access/). The ground-based lidar and radiosondes data can be obtained by contacting the corresponding

author.



*Authors contributions.* MR and SK conceived the study. SK, RW, MR and AH conducted the lidar measurements. AH, RW and PK offered scientific insight. JPC and YH conducted AboVE-Maido radiosonde measurements The paper is written by MR with contributions from all co-authors.

*Acknowledgements.* The upgrade of OHP and OPAR wind lidars was financially supported by CNES (Centre National d'Etudes Spatiales) as well as through EU FP7 ARISE and H2020 ARISE2 projects. We gratefully thank the personnel of OPAR station (Eric Gloubic, Patrick Hernandez and Louis Mottet) and the personnel of Station Gerard Megie at OHP (Frederic Gomez, Francois Dolon, Pierre Da Conceicao, Francois Huppert and others) for conducting the radiosonde launches and lidar operation. The work related to Aeolus validation has been performed in the frame of Aeolus Scientific Calibration & Validation Team (ACVT) activities

*Competing interests.* The authors claim no competing interests.

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
