# Peer review of "Validation of Aeolus wind profiles using ground-based lidar and radiosonde observations at La Réunion Island and the Observatoire de Haute Provence"

_EGUsphere, 2022_

## Referee Comment (RC3)

**Review report**

The topic of the submitted study is the evaluation of the Aeolus Rayleigh-clear winds against radiosondes and ground-based lidars acquired at two observatories (OHP, OPAR) in the framework of the AboVE validation campaigns. Moreover, an assessment throughout the satellite mission is performed using twice-daily routine Météo-France radiosondes and regular lidar observations. Overall, it is a very interesting work covering all the necessary aspects of a comprehensive Cal/Val study. I would like to acknowledge also that the authors are comparing their results against those obtained from numerous previous studies. However, I think that the weak point of the study is the absence of evaluation results for the Mie-cloudy winds. I believe that the authors should either support better their decision or include a similar analysis for the Mie-cloudy winds. Please find below my (minor) comments which should be addressed prior publishing the manuscript.

1. **Lines 32-33:** Please rephrase this sentence.
2. **Line 50:** Aeolus provides vertical profiles of HLOS and not of LOS.
3. **Line 56:** Replace "Aeolus's" with "Aeolus'".
4. **Lines 144-150:** Please explain why you are focusing only on Rayleigh-clear winds. The description of the cross-talk issue can be improved.
5. **Line 149:** It is the first time that the HLOS is mentioned in the text and should be written explicitly. Check all similar instances throughout the text.
6. **Lines 153-154:** Rephrase this sentence.
7. **Lines 164-170:** It will be helpful to mention here Figure 5a in Lux et al. (2020).
8. **Lines 188-194:** Can you add a figure visualizing the applied methodology? It is not clear to me why you are averaging the radiosondes measurements and the lidar retrievals between the Aeolus bins' middle points and not within their range (i.e., from base to top of each Aeolus bin).
9. **Lines 199-201:** Why the azimuth angles are the same between dawn and dusk Aeolus orbits?
10. **Lines 238-245:** Please consider rewriting and improving this paragraph. Can you explain better the statements "…*measurements better than 10km…*" and "…*still remained within 100km.*"? To my opinion, they are not obvious in the relevant figure.
11. **Lines 255 – 256:** This sentence needs a correction.
12. **Line 311:** How have you defined the 200 km window?
13. **Lines 317-319:** Please rephrase this sentence.
14. **Figure 3:** Do you see a different behavior when reproducing the same plots separately for each station?
15. **Line 347:** I think that you are referring to Figure 4.
16. **Line 412:** Which method?
17. **Lines 511-513:** I think that it is not feasible to generalize such results since maybe there are not valid for other stations characterized by different weather/wind regimes. There is also a similar statement in the Discussion section.
18. **Figure 6:** It would be very useful to use different shading colors (as background) corresponding to each baseline and show with double-edge arrows the two laser-periods (FM-A, FM-B).
19. **Lines 616-618 and lines 619-623:** There is a contradiction between these two parts. Can you clarify better your statements?

---

## Author Comment (AC1)

**Reply to Reviewer #1.**

**We thank Reviewer #1 for the positive review and fair remarks, which have all been carefully implemented in the manuscript.**

General comments:

I am very happy to read this article. It clearly outlines the data and methods used, and provides an important new result for the validation of the spaceborn Aeolus lidar.

Specific comments:

I am worried about the presentation of the overall statistics, which are an accumulation of Aeolus data for different baselines (for example in the abstact on lines 25,26).

The Aeolus instrument settings as well as the ground processing has seen several major changes during its mission. These will have an effect on statistical properties like bias and standard deviation/MAD. In addition to the combined statistics I think it would be better to split the results and also present them separately for the different baselines. Also it seems near-real-time and reprocessing results are mixed, i.e. baseline 11 was introduced in near-real-time processing on 8-Oct-2020, so the baseline 11 results before that date must be based on reprocessed Aeolus data. I think it would be better to split this as well, since the reprocessing used different calibration data than the near-real-time processing.

**Thank you for your suggestion to split the results and present them separately for different baselines. We appreciate your insight and will surely include this in our revised manuscript. We also appreciate your noting the mix of near-real-time and reprocessed results in the baseline 11 data, and we will ensure that this is clearly distinguished in the new Table 3.**

line 47: you state that Aeolus covers nearly the whole globe within 7 days.

This is not really the case. With a 7 day repeat cycle of the orbit the instrument observes a specific pattern on the earth and the slant curtain above this pattern, but it certainly does not observe every location on earth.

**After your comment, we decided to remove this line because we realized it didn't add any valuable information and generated confusion.**

line 453: The figure depicts a very specific pattern of oscillating nature.

This pattern is indeed striking, and I have not seen such a thing before in previous Aeolus publications. I think it is important to try and understand what is happening here. But I think you should not call this "instument induced", since you cannotиyet prove that this indeed is the case. There could also be some bug or unforeseen effect in the ground

processing or in the data handling of this paper. So I would suggest to find another name and not use the acronym I2OPs. Please contact the Aeolus DISC team and work with them to try and find what is happening here.

**You are correct that we cannot definitively prove that the observed pattern is instrument-induced. We have renamed the phenomenon "oscillating perturbations" instead of "instrument-induced oscillating perturbations."**

**We have also reached out to the Aeolus DISC team and are working with them to try and understand the cause of these oscillations (Here is the link to a confluence discussion about it: https://www.aeolus.esa.int/confluence/pages/viewpage.action?spaceKey=CALVAL&title=CC_CV_2B_015).**

Technical corrections:

line 13: Aeolus is now flying for over 4 years, so please correct your statement that it is operating for 3 years.

**We have corrected this imprecise statement.**

line 32: wind profiling crucial => wind profiling is crucial

**Corrected.**

line 49: the first ever Doppler-Rayleigh Wind Lidar =>the first ever Doppler-Rayleigh-Mie Wind Lidar

**Corrected.**

line 135: classified using particle backscatter coefficient

The classification method was changed to use SNR threshold on 31-Oct-2019 with the start of baseline 7 for the Mie channel, and on 8-Oct-2020 with the start if baseline 11 for the Rayleigh channel.

**Thank you for pointing out this error in the text. We apologize for the confusion caused by the change in the classification method. We have revised the manuscript to clearly state that the SNR threshold was introduced in 31-Oct-2019 for the Mie channel and on 8-Oct-2020 for the Rayleigh channel.**

line 159/160: the end of the mission's extended life in November 2022. ==>the end of the mission's extended life in spring 2023.

**Corrected.**

line 168: the difference between vLOS and HLOS becomes negligible

No this is not true. If w is small, than the sine term in equation (1) becomes negligible, but the cosine term stll remains. Therefore there still is a difference by a factor of cas(Psi) between vLOS and vHLOS.

**You are correct that the sine term in equation (1) becomes proportional to the cosine term when w is small, and we apologize for the error in our original manuscript. We have revised the manuscript to reflect this relationship accurately, and we have changed the wording from "negligible" to "proportional by a factor of cos(psi) ".**

line 181/182: multiple RBC settings are activated at the same time.

No this is not true. Each channel has just one RBS at any given time. But the RBS can be changed multiple times per orbit.

**You are correct that each channel has just one RBS at any given time. We have revised the manuscript to remove this sentence.**

line 192: The downsampling begins with an averaging of the reference measurements between the middle points of the reference bins

This phrasing is confusing and maybe I misunderstood.

Each Aeolus wind result has just one middle point, so there is no in between. So I think the correct way is to take the reference measurements between the top and bottom edge of the Aeolus measurement bin, and average these results, before comparing to the Aeolus result. That way no interpolation at all is needed.

**You are correct that the reference measurements are averaged between the top and bottom edge of the Aeolus measurement bin, rather than between the middle points of the reference bins. This is done to bring the reference measurements to the exact resolution as the Aeolus measurements. Our phrasing was incorrect, and we apologize for the confusion.**

**Additionally, you pointed out that each Aeolus wind result has just one middle point. This is true, but it also has a lower and higher bin bound. Therefore, when we reference the middle points of the reference bins, we are referring to the same thing as the top and bottom edge of the Aeolus measurement bin.**

**These changes have been reflected in the text : "Each Aeolus profile is used as a reference for the collocated profiles downsampling, meaning that the averaging grid is specific to each satellite observation. In order to match the resolution of the Aeolus measurements, we first average the reference measurements between the bounds of each Aeolus bin. This avoids the need for interpolation and ensures that**

**the reference measurements are at the same resolution as the Aeolus measurements."**

line 311: to average every profile => to average every Aeolus profile

**Corrected.**

line 345: shown in Fig. 5a => shown in Fig. 4a

**Corrected.**

line 387: One reason Sun et al. (2014) raised

One important contribution for orbital phase biases is the telescope temperature effect explained by Weiler et al., 2021.

I think you should mention this as well here.

**Thank you for your suggestion. We have indeed added the citation by Weiler et al., 2021 in our manuscript, and we appreciate your suggestion to include it.**

line 439: At the same time, the radiosonde drifts along

You could mention here that not only the distance between Aeolus and radiosonde changes with time, but also the time difference between the two systems changes with time and therefore also with altitude.

**We have added the following text to our manuscript: "Furthermore, not only the distance between Aeolus and radiosonde changes with time, but also the time difference between the two systems changes with time and, therefore, also with altitude."**

line 512: Once can thus conclude => One can thus conclude

**Corrected.**

line 530/531: the end of the extended mission lifetime in November 2022, =>the end of the extended mission lifetime in spring 2023,

**Corrected.**

line 580/581: there are fewer particles at higher altitude levels. =>there are fewer molecules at higher altitude levels.

**Corrected.**

line 603: higher by and average => higher by an average

**Corrected.**

---

## Author Comment (AC3)

**Reply to Reviewer #2.**

**We thank Reviewer #2 for the review and thorough work in finding issues and providing feedback on the readability of our manuscript. The efforts that will help us address any weaknesses in our manuscript are greatly appreciated, and we hope our revised manuscript meets the reviewer's expectations.**

In my opinion this is an excellent paper summarizing the comparison of an extensive data set of radiosonde and surface-based lidar observations with Aeolus wind estimates. The paper is quite comprehensive in its analysis of the data set. I especially like that the authors have compared their results with results from other groups carrying out similar investigations at different locations and using different instruments. This work is quite important for evaluating Aeolus performance and assessing the potential utility of the Aeolus observations.

Most of my comments are minor and editorial in nature, and I leave it to the authors and editor to decide on whether or not to include them in a revised manuscript.

Specific comments

Line 32: It seems that there should be an "is" inserted after "wind profiling".

**Corrected.**

Line 56: Perhaps I missed it, but it's a bit unclear whether the work described in the paper is investigating total bias or residual bias. As noted, bias correction schemes were implemented to the Aeolus data at different times in the mission. Although a fully bias-corrected data set is being developed for Aeolus, this analysis seems to be using data that may have included intermittent changes to the bias. A sentence or 2 to perhaps discuss bias correction in the Aeolus data set and which data are being used for comparison could be helpful to the reader.

**From our understanding, total bias is a measure of the overall accuracy of a measurement. In contrast, residual bias measures the remaining error after accounting for known sources of error. Both are important in understanding the reliability and accuracy of a measurement. However, the bias discussed in this paper is the residual bias, since it considers the many corrections put into place in the baselines, such as hot pixel and mirror temperature correction. Your remark considering intermittent changes to the bias is perfectly valid, and a new table (Table 3) has been added to reflect and document changes in the baselines, through the display of the residual bias and MAD for every given baseline, including reprocessed and real-time. We also added additional context in the text, where you suggested : "The Aeolus instrument settings and ground processing have significantly changed during its mission. These will affect statistical properties like bias and standard deviation/MAD. In addition to the combined statistics, we provide**

**in table 3 a splitting of the results, presenting them separately for the different baselines. Also, near-real-time and reprocessing results are separated, i.e., baseline 11 (introduced in near-real-time processing on 8-Oct-2020) and the baseline 11 results before that date (based on reprocessed Aeolus data). The split is needed since the reprocessing used different calibration data than the near-real-time processing."**

Line 79: Is LIOvent an acronym? If so, please define.

**LIOvent is not an acronym, it is just a name, similar to other lidar systems at OHP, e.g. LIO3S for the stratospheric ozone lidar.**

Line 93: Does the LIOvent lidar measure only Rayleigh winds?

**The LIOvent instrument senses the Doppler shift from both Rayleigh and Mie backscattering thanks to the spectral configuration of its double-edge FPI. Nevertheless, the Mie-type measurements are prone to larger error increasing with the backscatter ratio. The following sentence has been added: "The spectral configuration of the LIOvent FPI enables sensing the Doppler shift not only in clear air but also in the presence of thin clouds or aerosol layers, however the measurement error increases with the backscatter ratio (Souprayen et al., 1999)"**

Line 230, Figure 1 caption: I think that the caption could use a bit more explanation. Although I think that the straight colored trajectories represent lidar measurements from Aeolus and the ground-based lidar, this isn't described in the caption so it required some scrutiny on my part to discern what they represent. The same is true for the Figure 2 caption.

**After re-evaluating the description thanks to your comment, we decided to add another sentence to the description of the figures: "The straight colored strokes represent lidar measurements from Aeolus and the ground-based lidar".**

Line 244: Looks like a word is missing after "however".

**Corrected.**

Line 300 (Figure caption): It would be useful for the caption to differentiate between the lighter and darker shading and c and relate them (presumably) to the orbit.

**Thank you for your suggestion. We have taken your remark into account and have added the following sentence to the description: "The lines represent the average bias of each bin altitude, and the red (black) shading is the standard deviation of the bias in each range bin for ascending (descending) orbits."**

Line 319: I find the paragraphs beginning at line 305 and line 320 to be unclear. Does paragraph 305 refer to the radiosonde comparisons while 320 refers to the lidar

comparisons? Also the text beginning on line 305 refers to figures 3a, 3c, (wind measurement differences), while the text beginning on line 320 refers figures 3c(wind measurement differences) , 3d (data count), but seem to be doing an equivalent comparison.  These two paragraphs seem to require some clarification or correction.

**Thank you for pointing out this issue. We mistakenly said  "Figure 3a, 3c", where we meant "3a, 3b". Each paragraph should correspond to just one instrument and its data count, meaning it is a figure meant to be read horizontally, line by line. We hope that this small change helps clarify the paragraphs' arguments and their readability.**

Lines 345 and 347:  I think the authors mean "4a" instead of "5a" here.

**Yes, indeed, this issue was corrected.**

---

## Author Comment (AC4)

**Reply to Reviewer #3.**

**We thank Reviewer #3 for the appreciation of our work and the detailed review. All the remarks have been carefully addressed in the revised manuscript.**

The topic of the submitted study is the evaluation of the Aeolus Rayleigh-clear winds against radiosondes and ground-based lidars acquired at two observatories (OHP, OPAR) in the framework of the AboVE validation campaigns. Moreover, an assessment throughout the satellite mission is performed using twice-daily routine Météo-France radiosondes and regular lidar observations. Overall, it is a very interesting work covering all the necessary aspects of a comprehensive Cal/Val study. I would like to acknowledge also that the authors are comparing their results against those obtained from numerous previous studies. However, I think that the weak point of the study is the absence of evaluation results for the Mie-cloudy winds. I believe that the authors should either support better their decision or include a similar analysis for the Mie-cloudy winds. Please find below my (minor) comments which should be addressed prior publishing the manuscript.

1. Lines 32-33: Please rephrase this sentence.

**We rephrased the sentence to be "Therefore, continuous global wind profiling is essential for enhancing our understanding of atmospheric dynamics and improving the accuracy of numerical weather predictions (Houchi et al., 2010; Albertema et al., 2019; Stoffelen et al., 2005; 2020)"**

2. Line 50: Aeolus provides vertical profiles of HLOS and not of LOS.

**Corrected.**

3. Line 56: Replace "Aeolus's" with "Aeolus'".

**We corrected the three instances in the text where this error occurred.**

4. Lines 144-150: Please explain why you are focusing only on Rayleigh-clear winds.

The description of the cross-talk issue can be improved.

**The study focuses on the Rayleigh-clear wind cal/val for the following reasons.**

**First, both ground-based DWLs only have one detection channel based on the double-edge FPI, that is the same as the ALADIN Rayleigh channel. Thanks to the spectral configuration of the ground-based lidars' FPI, the measurements of Doppler shift using the Mie scattering are possible within thin cirrus clouds and aerosol layers, however the measurement error increases with the backscatter ratio. The respective mention has been included into the DWL description in the manuscript. In addition, both lidars are optimized for the middle atmosphere and**

**cannot measure winds within the boundary layer, where the aerosols are more abundant.**

**Second, for the above reason, the DWL measurement sessions (and collocated radiosoundings) were mostly restricted to the clear-sky conditions, which substantially limited the number of the collocated Mie detections by ALADIN. Occasional high-level thin cirrus clouds, occurring during the measurement sessions, do not allow for drawing up conclusive intercomparison statistics.**

**A sentence has been added to the introduction: "Since the optimal performance of the ground based Doppler lidars is achieved in the clear sky conditions, this paper will only focus on the ALADIN Rayleigh clear data analysis. Rayleigh clear stands for clear skies."**

**We do consider performing a separate study focusing on the Mie cal/val that will take advantage of the Hunga Tonga stratospheric aerosols that have been extensively sampled over La Reunion since January 2022.**

**A sentence has been added to the discussion: "With this study, we have addressed the performance of the ALADIN Rayleigh channel at a broad range of altitudes, from the lower troposphere to the maximum altitude of 30 km enabled by the AboVE-2 range bin setting. The performance of the ALADIN Mie channel in the lower stratosphere remains to be assessed using the lidar and radiosonde measurements at La Reunion. This site was to provide the most extensive lidar observations of the 2022 Hunga Tonga volcanic eruption plumes in the stratosphere (Baron et al., 2022), that were sampled by the ALADIN Mie channels (Legras et al., 2022; Khaykin et al., 2022)."**

5. Line 149: It is the first time that the HLOS is mentioned in the text and should be written explicitly. Check all similar instances throughout the text.

**This issue has been corrected since HLOS is now mentioned in line 56.**

 6. Lines 153-154: Rephrase this sentence.

**We rephrased the sentence to be "In the following study, we present data from baselines ranging from 2B02 and from 2B11 to 2B13, covering the period from September 2018 to January 2022."**

7. Lines 164-170: It will be helpful to mention here Figure 5a in Lux et al. (2020).

**Added a reference to (Lux et al., 2020a, their fig. 5a).**

8. Lines 188-194: Can you add a figure visualizing the applied methodology? It is not clear to me why you are averaging the radiosondes measurements and the lidar

retrievals between the Aeolus bins' middle points and not within their range (i.e., from base to top of each Aeolus bin).

**Thank you for your suggestion. We apologize if our original explanation was not clear. We have already addressed this issue in a previous comment, where we explained that the reference measurements are averaged between the top and bottom edge of the Aeolus measurement bin, rather than between the middle points of the reference bins.**

**Each Aeolus profile serves as a reference for the downsampling of collocated profiles, meaning that the downsampling grid is specific to each satellite observation. The downsampling procedure involves averaging the reference measurements between each Aeolus bin bound (which is the same as saying "the averaging window being half the distance between the upper and lower adjacent bins"). This allows the reference measurements to be brought to the exact resolution as the Aeolus measurements, without the need for interpolation.**

9. Lines 199-201: Why the azimuth angles are the same between dawn and dusk Aeolus orbits?

**The values displayed were only the ones corresponding to the ascending orbit. The corresponding values for descending orbits have been added. The text now reads :**

**" Where θ (259.9°/100° for OHP and 259.0°/101° for Maido, for ascending/descending orbits) is the topocentric azimuth angle, which is defined clockwise from north of the horizontal projection of the target to the satellite pointing vector. Therefore, each observation site has its own azimuth angle value."**

10. Lines 238-245: Please consider rewriting and improving this paragraph. Can you explain better the statements "…measurements better than 10km…" and "…still remained within 100km."? To my opinion, they are not obvious in the relevant figure.

**We apologize for the error in the previous text. We mistakenly wrote 100 km instead of 200 km. The correct statement is that the ascending orbit remained within a distance of 200 km after the ANX configuration was changed to ANX 2.0. The new text is as follows:**

**"The ANX, or Ascending Node crossing, is the point where the orbit of Aeolus intersects the x-y plane in the Earth's fixed coordinate system. During the campaign, the orbit parameter for the ANX was changed from ANX 4.5 to ANX 2.0 (as shown in Fig. 1) to support the Aeolus tropical campaign activities in Cape Verde. This change resulted in a shift in the orbit's location relative to the**

**observatory. Previously, the ANX 4.5 ascending orbit was located within 10 km of the lidar's eastward line-of-sight in the lower stratosphere on Wednesdays. After this change, the ascending orbit moved further away from the lidar's eastward line of sight, but remained within a distance of 200 km."**

11. Lines 255 – 256: This sentence needs a correction.

**The sentence was corrected to "During both campaigns, 19 Aeolus-collocated RS ascends were carried out, and 15 were time-coordinated with ground-based lidar acquisitions."**

12. Line 311: How have you defined the 200 km window?

**The collocation window of 200 km was chosen empirically as a trade-off between the number of collocations and their proximity. The goal was always to obtain around 2 or 3 different profiles, but no more. This approach allows for a good balance between having enough profiles to get an accurate average, while still avoiding any outliers that might skew the results.**

13. Lines 317-319: Please rephrase this sentence.

**We replaced the sentence with "The AboVE OHP2 lidar measurements were the only ones that had extended coverage below 5 km, which significantly reduced the number of data points in the lower troposphere."**

14. Figure 3: Do you see a different behavior when reproducing the same plots separately for each station?

**After conducting additional analysis, we did not notice any notable differences when reproducing the same plots separately for each station. While we are confident in the accuracy of our results, it is always important to consider the possibility of variations within the data.**

15. Line 347: I think that you are referring to Figure 4.

**Corrected.**

16. Line 412: Which method?

**Corrected to say "both methods".**

17. Lines 511-513: I think that it is not feasible to generalize such results since maybe there are not valid for other stations characterized by different weather/wind regimes. There is also a similar statement in the Discussion section.

We agree with your opinion that it is not feasible to generalize the results of this study to other stations characterized by different weather and wind regimes. You are correct that the results of this study may not be directly applicable to all stations, as the characteristics of different stations can vary significantly.

However, based on the results of this study, it is possible to conclude that temporal offset is more critical than spatial offset when collocating satellite and ground-based wind measurements, at least in the specific context of this study. This conclusion is based on the observed higher random error at the site with a more significant time offset, and similar patterns may be observed at other locations with similar characteristics.

Therefore, rather than making a generalization about all stations, it may be more appropriate to focus on the specific context of this study and the conclusions that can be drawn from the results within that context, as you suggest.

Here is an additional sentence we provided for context: "While it might not be trivial to generalize these results to other stations with different weather and wind regimes, the findings of this study may be relevant to locations with similar characteristics."

The discussion was also modified.

18. Figure 6: It would be very useful to use different shading colors (as background) corresponding to each baseline and show with double-edge arrows the two later periods (FM-A, FM-B).

Your remarks were added into the revised version of the figure, including both a varying coloring depending on the baseline and arrows annotations to provide context on the periods. In addition, we added this short sentence in the figure description :" The black line represents the average value, and the shading represents its standard deviation. The colors are relative to the 4 baselines used: 2B02(violet), 2B11(blue), 2B12(green) and 2B13(yellow) in that order."

19. Lines 616-618 and lines 619-623: There is a contradiction between these two parts. Can you clarify better your statements?

The first half of the text states that the current study did not observe any significant difference between the ascending and descending phases. This goes against previous observations that there are orbit-dependent characteristics. The second half of the text presents the results of the current study, which show that the mean correlation coefficients and scaled MAD values for the ascending and descending

phases are similar. Therefore, the first half of the text does not contradict the second half because the current study's results do not support the idea of orbit-dependent characteristics, as previously observed.

One potential explanation for the similarity in the results could be that the atmospheric conditions during the ascending and descending phases were similar, leading to similar measurements. This could be due to meteorological phenomena such as inversion layers, which can cause temperature and moisture profiles to be relatively stable over a specific altitude range. Additionally, the similarity in the results could be due to the accuracy and precision of the instrumentation, which has been calibrated to minimize any differences between the ascending and descending phases.

---

## Referee Report (RR1)

General comments:

in general I am pleased with the response of the authors to my previous review comments and the way they improved the paper in response of this.
I have only a few small minor comments on this new version.

specific comments:

line 146: in the lines above some explanation was added to the changes that happened in the classification algorithm, especially the change in using a threshold on scattering ratio to using a threshold in Mie SNR.
However on line 146 you state that "The method currently applied by ESA is
to use the scattering ratio" and this is not correct.
Currently a threshold on Mie SNR is used for classification of the Rayleigh channel. Only before baseline 2B11 the Rayleigh channel applied a threshold on the scattering ratio as derived from the Mie channel.
So please correct this.

line 162: You write "except we do not apply any HLOS error threshold."
But it is not entirely clear to me what you intent to say here. Do you mean you do not apply a threshold check on the estimated error? Or does this refer to a check on the difference between Aeolus wind and reference NWP wind?

line 306: here you write: "Also, near-real-time and reprocessing results are
separated"
I think it may not be obvious for the reader what the main differences are between both periods for baseline 11.
The main improvements I think are that the hot-pixel correction has been improved upon by also carefully considering the steps that occurred between the DUDE calibrations. This should mitigate the problems created by hot pixels.
Also the M1 telescope temperature correction procedure was applied in a different way, using data of the day itself rather than data of the day before to tune the correction parameters. This should clearly improve the overall and local biases.
See: https://earth.esa.int/eogateway/documents/20142/0/Aeolus-Summary-Reprocessing-2-DISC.pdf

line 525 Figure 6
I really like this new figure, clearly giving the periods when the different baselines were applied. Very nice to see also that the change between the reprocessed and NRT period of baseline 11 is very small.
Just one suggestion would be to also provide the bias results in a similar way.
It would be interesting to see how these changed (or not) with time.

---

## Author Response (AR2)

**Reply to Reviewer #1.**
**We thank, once again, Reviewer #1 for the positive review and thoroughness, which have helped this paper attaint a better level of quality.**

General comments: in general I am pleased with the response of the authors to my previous review comments and the way they improved the paper in response of this. I have only a few small minor comments on this new version.

specific comments:

line 146: in the lines above some explanation was added to the changes that happened in the classification algorithm, especially the change in using a threshold on scattering ratio to using a threshold in Mie SNR.

However on line 146 you state that "The method currently applied by ESA is to use the scattering ratio" and this is not correct. Currently a threshold on Mie SNR is used for classification of the Rayleigh channel. Only before baseline 2B11 the Rayleigh channel applied a threshold on the scattering ratio as derived from the Mie channel.

So please correct this.

**We indeed forgot to change subsequently the rest of the paragraph, creating a confusion. Thanks to your observation, it has now been changed to "The method currently applied by ESA is to use the Mie SNR threshold for classification of the Rayleigh channel ". The following scattering ratio mentions have also been changed to SNR.**

line 162: You write "except we do not apply any HLOS error threshold." But it is not entirely clear to me what you intent to say here. Do you mean you do not apply a threshold check on the estimated error? Or does this refer to a check on the difference between Aeolus wind and reference NWP wind?

**We are referring to the estimated error, which is the value Aeolus self-diagnoses concerning his own uncertainty. We added "[…] except we do not apply any HLOS estimated error threshold."**

line 306: here you write: "Also, near-real-time and reprocessing results are separated" I think it may not be obvious for the reader what the main differences are between both periods for baseline 11. The main improvements I think are that the hot-pixel correction has been improved upon by also carefully considering the steps that occurred between the DUDE calibrations. This should mitigate the problems created by hot pixels. Also the M1 telescope temperature correction procedure was applied in a different way, using data of the day itself rather than data of the day before to tune the correction parameters. This should clearly improve the overall and local biases. See: https://earth.esa.int/eogateway/documents/20142/0/Aeolus-Summary-Reprocessing-2-DISC.pdf

**We thank the reviewer for sharing this very interesting and synthetic report. We decided to add a reference to the document, along with a short explanation in the text: "The split is needed since the reprocessing used different calibration data than the near-real-time processing, along with several changes in the correction. The main refinements in the reprocessed dataset are improvements in the hot pixel correction by a more careful application of the method, along with using the data of the day itself rather than the day before for the M1**

**telescope temperature correction. The newer baseline should expect improved overall and local biases (ESA., 2021b)."**

line 525 Figure 6 I really like this new figure, clearly giving the periods when the different baselines were applied. Very nice to see also that the change between the reprocessed and NRT period of baseline 11 is very small. Just one suggestion would be to also provide the bias results in a similar way. It would be interesting to see how these changed (or not) with time

**Thank you for your feedback on Figure 6. We are glad that you found it informative. We have taken your suggestion into consideration and have added a similar figure displaying the bias results over time. This helps to show the changes in bias, if any, along with the different baselines applied. We added this description as well: "In addition to the standard deviation observed in Fig. 6, the bias can offer another opportunity to assess the evolution of the satellite's performance over time. The evolution of the bias shows a structure very similar to the previous figure, reducing its variability along the newer baselines. The expected higher variability for the OHP site is also observed, and the average value tends to be slightly lower ( -1 ms-1 for OHP to 0 ms-1 for Maido) in the latest 2B13 Baseline (December 2021). This figure suggests that the newer baselines help the bias converge to zero but do not have a definitive impact on the variability of the values. Additionally, the transition from reprocessed to real-time reprocessed data, which occurred on the 8th of October 2020 (ESA., 2021b), does not offer any apparent enhancements. Therefore, it does not support the assumption of any beneficial or detrimental changes to the data quality."**